# Customizing the Inductive Biases of Softmax Attention using Structured Matrices

**Yilun Kuang** [1]  **Noah Amsel** [1]  **Sanae Lotfi** [1]  **Shikai Qiu** [1]  **Andres Potapczynski** [1]  **Andrew Gordon Wilson** [1]

## Abstract

The core component of attention is the scoring function, which transforms the inputs into low-dimensional queries and keys and takes the dot product of each pair. While the low-dimensional projection improves efficiency, it causes information loss for certain tasks that have intrinsically high-dimensional inputs. Additionally, attention uses the same scoring function for all input pairs, without imposing a distance-dependent compute bias for neighboring tokens in the sequence. In this work, we address these shortcomings by proposing new scoring functions based on computationally efficient structured matrices with high ranks, including Block Tensor-Train (BTT) and Multi-Level Low Rank (MLR) matrices. On in-context regression tasks with high-dimensional inputs, our proposed scoring functions outperform standard attention for any fixed compute budget. On language modeling, a task that exhibits locality patterns, our MLR-based attention method achieves improved scaling laws compared to both standard attention and variants of sliding window attention. Additionally, we show that both BTT and MLR fall under a broader family of efficient structured matrices capable of encoding either full-rank or distance-dependent compute biases, thereby addressing significant shortcomings of standard attention.

## 1. Introduction

The attention mechanism (Bahdanau et al., 2016; Vaswani et al., 2017) is crucial to much of contemporary deep learning, providing a powerful and flexible way to process sequences. Because of its widespread application to so many different architectures and domains, it is common to think of attention as a general purpose tool (Bommasani et al., 2022). However, attention endows models with a certain set of inductive biases that suit some tasks better than others (Lavie et al., 2024). Moreover, attention is computationally expensive in both runtime and memory. A large ongoing research effort aims to devise more efficient alternatives to standard attention, especially for long sequences and big models (Katharopoulos et al., 2020; Beltagy et al., 2020; Gu & Dao, 2024; Dao & Gu, 2024; Yuan et al., 2025). Architectures like linear attention based Transformers come with their own sets of inductive biases that can be too limiting for real data and often reduce accuracy (Arora et al., 2024; Jelassi et al., 2024). In this work, we show how structured matrices can be used to customize attention, changing its inductive biases to suit particular tasks and thereby improving efficiency.

We focus on the attention scoring function, which determines how much each token attends to every other token by taking a dot product of their corresponding query and key vectors. We identify two properties of the scoring function that are suboptimal in certain settings. First, it can suffer from a *low-rank bottleneck* (Amsel et al., 2024). Since the head dimension, which defines the size of queries and keys, is significantly smaller than the embedding dimension, transforming inputs into queries and keys can cause information about each token to be lost. This bottleneck can limit the power of the attention layer. In fact, there are simple functions that are easy to express with attention if the head dimension is large enough, but are provably hard to express otherwise (Amsel et al., 2024).

Second, the form of the scoring function lacks an inductive bias for encoding distance-dependent compute. In standard attention, every pair of tokens is scored using the same function and the same weights, and each token can attend to the entire context. This feature makes attention powerful, but for long contexts, it is also expensive. Real-world datasets often exhibit some amount of locality, meaning that the semantics of a token depends most strongly on nearby tokens. Previous efforts have been made to make attention more efficient by promoting locality while still preserving some global communication. For instance, the Longformer architecture combines sliding window attention with a few

[1]New York University. Correspondence to: Yilun Kuang <yilun.kuang@nyu.edu>, Andrew Gordon Wilson <andrewgw@cims.nyu.edu>.

*Proceedings of the 42nd International Conference on Machine Learning*, Vancouver, Canada. PMLR 267, 2025. Copyright 2025 by the author(s).

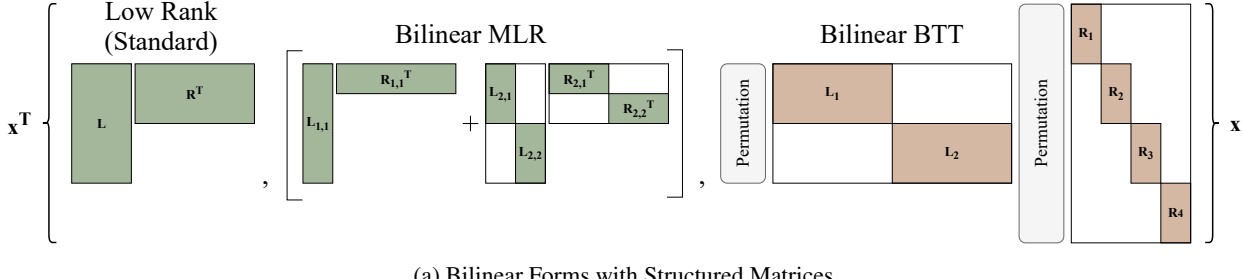

(a) Bilinear Forms with Structured Matrices

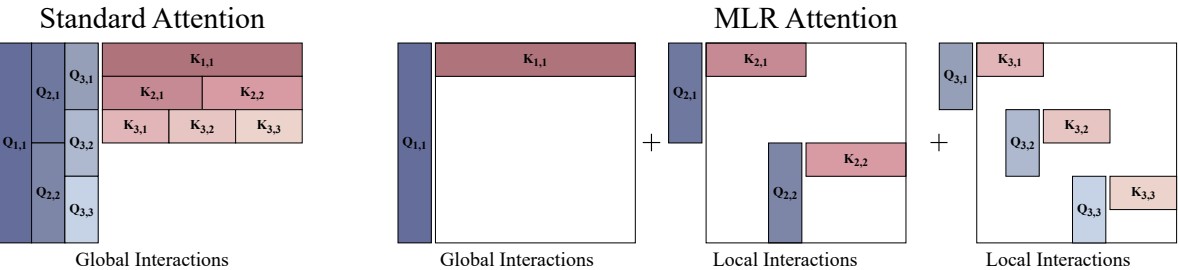

(b) Encoding Distance-Dependent Compute Bias with Multi-Level Low Rank (MLR) Attention

*Figure 1.* **Overview of two ways to customize the inductive biases of softmax attention with structured matrices.** (a) Standard attention computes the dot product between a query and a key via a low-rank bilinear transformation. In this work, we replace the low rank product with other structured matrices such as Multi-Level Low Rank (MLR) and Block Tensor Train (BTT) as introduced in Section 3.2. (b) Standard attention captures pair-wise token interactions in the global context without imposing a distance-dependent compute bias. We introduce an inductive bias for more flexible interactions between nearby tokens by changing the attention score matrix from low rank to Multi-Level Low Rank (MLR), as in Section 3.4.

tokens of standard attention (Beltagy et al., 2020). However, such constructions are often brittle and gain efficiency at the expense of accuracy (Zhang et al., 2024).

To address these limitations, we introduce rich classes of structured matrices into the attention scoring function. Selecting an appropriate structure allows us to refine the inductive bias introduced by attention, tailoring it to specific tasks. Prior work has demonstrated the advantages of replacing dense linear layers with structured matrices (Dao et al., 2022; Qiu et al., 2024; Potapczynski et al., 2024), but these approaches treat each linear layer separately. Attention scores are built from a combination of multiple linear and bilinear transformations. By replacing all of these transformations with a single structured matrix, we expand the design space for more efficient and expressive attention mechanisms.

We explore two applications of this approach. To address the low-rank bottleneck, we incorporate Block Tensor Train (BTT) and Multi-Level Low Rank (MLR) matrices, structured families that have high rank while remaining efficient in terms of parameters and computation. Using this technique, we demonstrate improvements for in-context linear regression, a task where standard attention is constrained by the low-rank bottleneck. To better allocate compute and memory resources, we apply MLR matrices in a different

way that prioritizes local interactions over long-range dependencies. This leads to mild improvements for language modeling compared to standard and sliding window attention and for time series forecasting with long horizons.

Our contributions are:

- We introduce a conceptual framework for analyzing and modifying the inductive biases of attention through the structure of its underlying linear and (bi-)linear transformations.

- In Section 3.2 and Section 4, we apply this framework to eliminate the low-rank bottleneck of standard attention using high-rank BTT and MLR matrices, improving performance on an inherently high-dimensional task from the literature (Garg et al., 2022).

- In Section 3.3, we show that both BTT and MLR matrices—including Monarch (Dao et al., 2022), Butterfly (Dao et al., 2020), Kronecker, and Low Rank matrices—can be united under a broader structured family which we call Multi-Level Block Tensor Contraction (MLBTC).

- In Sections 3.4 and 5, we use MLR matrices to introduce a distance-dependent compute bias, which

slightly outperforms previous methods in language modeling and time series forecasting.

This work advances our understanding of attention's inductive biases, exploring its structural limitations and offering a principled approach to design more efficient and expressive architectures.

Our codes are available at the following github repository https://github.com/YilunKuang/structured-attention.

# 2. Inductive Biases and Limitations of Standard Attention

In the following section, we discuss how the built-in inductive biases of attention can be inappropriate in certain settings. In Section 2.1, we introduce our notations and review standard multi-head attention (Vaswani et al., 2017). We identify the scoring function for sequence mixing as a bilinear transformation. In Section 2.2, we show that the scoring function in multi-head attention relies on a low rank matrix, which creates information bottleneck for tasks with intrinsically high dimensional inputs. On in-context regression tasks, we show that multi-head attention requires a large enough head dimension to achieve good regression performance. In Section 2.3, we point out that standard attention shares the same scoring function for all tokens within the context without imposing a distance-dependent compute bias. Real data often exhibits locality patterns, and thus attention fails to exploit the structure of data for improved efficiency.

## 2.1. The Attention Scoring Function

Let $H$ be the number of attention heads, $D$ be the embedding dimension, and $r$ be the head dimension, where $D = Hr$. The input to a multi-head self-attention layer is a $T \times D$ matrix $\mathbf{X}$, where $T$ is the sequence length. The output is a $T \times D$ matrix given by

$$\sum_{i=1}^{H} \sigma \left( \mathbf{X} \mathbf{W}_{Q_i} \mathbf{W}_{K_i}^{\top} \mathbf{X}^{\top} \right) \mathbf{X} \left( \mathbf{W}_{V_i} \mathbf{W}_{O_i}^{\top} \right) \qquad (1)$$

where $\sigma$ is the row-wise softmax function and $\mathbf{W}_{Q_i}, \mathbf{W}_{K_i}, \mathbf{W}_{V_i}, \mathbf{W}_{O_i} \in \mathbb{R}^{D \times r}$ are weight matrices. Attention is usually described with reference to queries, keys and values (given by $\mathbf{X} \mathbf{W}_{Q_i}$, $\mathbf{X} \mathbf{W}_{K_i}$, and $\mathbf{X} \mathbf{W}_{V_i}$) but in this paper we take a different perspective. Consider any head and define its attention scoring function $s : \mathbb{R}^D \times \mathbb{R}^D \to \mathbb{R}$ as follows:

$$s(\mathbf{x}, \mathbf{x}') = \mathbf{x}^{\top} \mathbf{W}_{Q_i} \mathbf{W}_{K_i}^{\top} \mathbf{x}' \qquad (2)$$

This is simply the bilinear form given by the matrix $\mathbf{W}_{Q_i} \mathbf{W}_{K_i}^{\top}$. We define the score matrix $\mathbf{S} \in \mathbb{R}^{T \times T}$ by $\mathbf{S}_{j,j'} = s(\mathbf{x}_j, \mathbf{x}_{j'})$. Then the output of this head can be

rewritten as follows:

$$\sigma \left( \mathbf{S} \right) \mathbf{X} \left( \mathbf{W}_{V_i} \mathbf{W}_{O_i}^{\top} \right) \qquad (3)$$

In this paper, we explore alternatives to the standard scoring function given in Equation (2). We now describe two of its drawbacks.

## 2.2. Low-Rank Bottleneck

Any $D \times D$ matrix defines a bilinear form, so in principle any $D \times D$ matrix could be used in Equation (2) to define an attention scoring function. Some earlier forms of attention used a single $D \times D$ weight matrix (Luong et al., 2015), but Vaswani et al. (2017) chose to use the rank-$r$ matrix $\mathbf{W}_{Q_i} \mathbf{W}_{K_i}^{\top}$ instead. Since $r \ll D$, this low-rank scoring function is more efficient than a full-rank one, but it also has weaker expressive power. For example, a natural scoring function for many tasks is $s(\mathbf{x}, \mathbf{x}') = \mathbf{x}^{\top} \mathbf{x}'$, but a low-rank matrix is not capable of representing it. This phenomenon is called the low-rank bottleneck of multi-head attention.

Recent work has shown that the low-rank bottleneck can seriously weaken attention-based models in certain settings. Amsel et al. (2024) demonstrated that for in-context linear regression, a task popularized by Garg et al. (2022), one full-rank attention head can significantly outperform low-rank multi-head attention after controlling for model size. They also proved that an attention layer cannot solve the nearest-neighbor problem for points on the $d_{\text{input}}$-dimensional sphere, even approximately, unless $r \gtrsim d_{\text{input}}$. In both cases, the inputs are intrinsically high-dimensional and cannot be compressed without losing important information. In such settings, the efficiency of low-rank attention comes at the price of accuracy.

In Figure 2, we demonstrate this unfavorable trade-off for in-context regression, replicating the setting from Garg et al. (2022). Our results show that, across a range of model sizes, transformers cannot solve this task unless the head dimension $r$ is nearly as large as the input dimension. Once this threshold is reached, transformers achieve high accuracy. In Section 3.2, we explore alternatives to the low-rank scoring function of Equation (2) that are both full-rank and efficient.

## 2.3. Distance-Dependent Compute Bias

An important feature of attention-based models is that they allow any pair of tokens to exchange information. A standard attention layer is completely agnostic to the order of its inputs. In particular, the same scoring function $s(\cdot, \cdot)$ is used for every pair of tokens, regardless of where they appear in the sequence.

Many real-world tasks exhibit locality patterns, meaning that tokens appearing near each other in a sequence are more likely to be connected. In natural language, for instance,

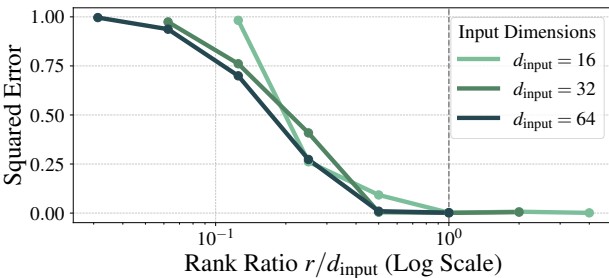

*Figure 2.* **Multi-head attention cannot solve in-context regression unless the head dimension $r$ is close to the input dimension** $d_{\text{input}}$. We train 6-layer transformers with 8 heads and varying embedding and head dimensions to perform in-context regression: given the prompt $\mathbf{x}_1, \mathbf{w}^\top \mathbf{x}_1, \ldots, \mathbf{x}_{N-1}, \mathbf{w}^\top \mathbf{x}_{N-1}, \mathbf{x}_N$ for $\mathbf{x}_i \in \mathbb{R}^{d_{\text{input}}}$ and unknown $\mathbf{w}$, predict $\mathbf{w}^\top \mathbf{x}_N$. Plot shows error at $N = 2 d_{\text{input}}$. For details see Section 4.

words appearing in the same sentence or paragraph must be understood together, while words appearing hundreds of pages apart usually do not influence each other directly. There are exceptions, such as when a particular fact must be recalled across a long context. Attention does not naturally exploit the local patterns in the data, so it cannot take advantage of this simplifying structure. For autoregressive models, the global nature of attention also requires the keys and values of all previous tokens to be saved in cache, which is expensive.

Due to these drawbacks, previous work has altered attention by imposing a sparsity pattern on the attention score matrix $\mathbf{S}$. For instance, sliding window attention (SWA) masks pairs of tokens that are more than some fixed distance apart (Child et al., 2019). In effect, this changes the score matrix to use different scoring functions for different pairs of tokens:

$$s_{j,j'}(\mathbf{x}_j, \mathbf{x}_{j'}) = \begin{cases} \mathbf{x}_j^\top \mathbf{W}_Q \mathbf{W}_K^\top \mathbf{x}_{j'} & \text{if } |j - j'| \le T'. \\ -\infty & \text{otherwise.} \end{cases}$$

where $T'$ is the window size and we drop the attention head index $i$ for ease of notation. Some models use a strided version of SWA or give a global receptive field to a few tokens (Beltagy et al., 2020). However, these sparse versions of attention are brittle and underperform standard attention (Arora et al., 2024), so they are commonly combined with some form of non-sparse attention (Riviere et al., 2024; Warner et al., 2024; Behrouz et al., 2024b). In Section 3.4, we introduce a flexible and hierarchical distance-dependent compute bias by making $\mathbf{S}$ a structured (but not sparse) matrix.

# 3. Customizing the Scoring Function with Structured Matrices

In this section, we show how to modify the scoring function so as to change its inductive biases and alleviate the issues identified in the previous section. We first introduce our main mathematical tool, structured matrix families, in Section 3.1. We then show two ways to incorporate these structured matrices into the scoring function, addressing the issues identified in Sections 2.2 and 2.3. In addition, in Section 3.3 we identify a generalization of the existing structured matrix families. Finally, we cover practical considerations regarding efficient implementations of structured matrices and techniques for stable feature learning in Section 3.5.

## 3.1. Structured Matrix Families

Consider $D \times D$ matrices. A structured matrix is one that can be represented using fewer than $D^2$ parameters. Structured matrices often admit fast algorithms for matrix-vector multiplication, making them both computationally- and parameter-efficient. For instance, **low rank** matrices have the form $\mathbf{LR}^\top$, where $\mathbf{L}, \mathbf{R} \in \mathbb{R}^{D \times r}$ and $r \ll D$. They require only $2rD$ parameters to represent, and the corresponding bilinear form $(\mathbf{x}, \mathbf{y}) \mapsto \mathbf{x}^\top \mathbf{LR}^\top \mathbf{y}$ can be computed with only $O(rD)$ FLOPs. We consider several other families of structured matrices in this paper. Key properties of each class are summarized in Table 1. We also present the extension to structured matrices with rectangular shapes in Appendix A.

**Block diagonal** matrices contain all zeros except for $p$ blocks along the diagonal, which are each dense. We notate this as $\bigoplus_{k=1}^p \mathbf{W}_k$, where each $\mathbf{W}_k$ is a $\frac{D}{p} \times \frac{D}{p}$ diagonal block and $\bigoplus$ denotes the direct sum. In a **low-rank block diagonal** matrix, each of these diagonal blocks is itself low rank. They have the form $\bigoplus_{k=1}^p \mathbf{L}_k \mathbf{R}_k^\top$, where $\mathbf{L}_k$ and $\mathbf{R}_k \in \mathbb{R}^{\frac{D}{p} \times r}$. Intuitively, this family interpolates between low-rank matrices ($p = 1$) and diagonal matrices ($p = D$).

**Multi-Level Low Rank (MLR)** matrices (Parshakova et al., 2023; 2024) are summations of several low-rank block diagonal matrices with different block sizes. Formally, an $L$-level MLR matrix is given by

$$\sum_{l=1}^L \bigoplus_{k=1}^{p_l} \mathbf{L}_{l,k} \mathbf{R}_{l,k}^\top \tag{4}$$

where $\mathbf{L}_{l,k}$ and $\mathbf{R}_{l,k}$ are $\frac{D}{p_l} \times r_l$ low-rank factors. Each level represents interactions at a particular scale. MLR matrices with a range of block sizes inherit the abilities of both low-rank ($p_l = 1$) *and* block diagonal matrices ($p_l \gg 1$), simultaneously capturing coarse-grained global interactions and fine-grained local interactions. Thus, they

*Table 1.* **Properties of Structured Matrix Families**. For all listed structures, the number of FLOPs needed to compute the bilinear form $\mathbf{x}^\top \mathbf{M} \mathbf{y}$ is $O(\#\text{params})$. For BTT, we assume $a = b = c = d = \sqrt{D}$. MLR and BTT both attain high rank relative to their parameter counts.

| Structure | Definition | Parameters | Rank |
|---|---|---|---|
| Dense | $\mathbf{W}$ | $D^2$ | $D$ |
| Low Rank | $\mathbf{L}\mathbf{R}^\top$ | $2Dr$ | $r$ |
| Multi-Level Low Rank (MLR) | $\sum_{l=1}^{L} \bigoplus_{k=1}^{p_l} \mathbf{L}_{l,k}\mathbf{R}_{l,k}^\top$ | $2D\sum_l r_l$ | $\sum_l r_l p_l$ |
| Block Tensor Train (BTT) | $\mathbf{P}_L(\bigoplus_{k'=1}^{b} \mathbf{L}_{k'})\mathbf{P}_R(\bigoplus_{k=1}^{b} \mathbf{R}_k^\top)$ | $2D^{\frac{3}{2}}s$ | $D$ |

are a natural and efficient way to represent hierarchical structure.

**Block Tensor Train (BTT)** matrices were introduced in Qiu et al. (2024) as a generalization of Monarch (Dao et al., 2022) and Butterfly (Dao et al., 2020) matrices. They are designed to be efficient, expressive, and full-rank. Qiu et al. (2024) and Potapczynski et al. (2024) found that they outperform other structured matrix families as replacements for linear layers in neural networks. Given hyperparameters $a, b, c, d$ and $s$, where $ab = cd = D$, a BTT matrix has the form

$$\mathbf{P}_L \left( \bigoplus_{k'=1}^{b} \mathbf{L}_{k'} \right) \mathbf{P}_R \left( \bigoplus_{k=1}^{c} \mathbf{R}_k^\top \right) \quad (5)$$

where $\mathbf{L}_{k'} \in \mathbb{R}^{a \times cs}$, $\mathbf{R}_k \in \mathbb{R}^{d \times bs}$ and $\mathbf{P}_L \in \mathbb{R}^{D \times D}$, $\mathbf{P}_R \in \mathbb{R}^{cbs \times cbs}$ are fixed permutation matrices. $\mathbf{P}_L$ permutes the rows by rearranging the dimension $b \cdot a$ into $a \cdot b$. This permutation is equivalent to reshaping a vector $\mathbf{z} \in \mathbb{R}^{ba}$ into a matrix $\mathbf{Z} \in \mathbb{R}^{b \times a}$, transposing it to $\mathbf{Z}^\top \in \mathbb{R}^{a \times b}$, and vectorizing it to get $\mathbf{z}' \in \mathbb{R}^{ab}$. Likewise, $\mathbf{P}_R$ reshapes a vector of dimension $c \cdot b \cdot s$ into a tensor with shape $(c, b, s)$, swaps the first two dimensions, then flattens it back into a vector of dimension $b \cdot c \cdot s$. Qiu et al. (2024) proved that when $a = b = c = d = s = \sqrt{D}$, BTT can express any $D \times D$ matrix. For efficiency, we set $s = 1$ or $s = 2$.

### 3.2. Resolving the Low-Rank Bottleneck with Structured Bilinear Forms

As described in Section 2.2, parameterizing the attention scoring function by a low-rank matrix $\mathbf{W}_Q \mathbf{W}_K^\top$ creates a information bottleneck. However, using a dense $D \times D$ matrix is prohibitively expensive; evaluating the scoring function would require $O(D^2)$ operations per attention head. Instead, we propose using a structured matrix that is both high-rank and allows efficient evaluation of the scoring function. In particular, we use MLR and BTT matrices as

visualized in Figure 1a:

$$s_{\text{MLR}}(\mathbf{x}_j, \mathbf{x}_{j'}) = \mathbf{x}_j^\top \left( \sum_{l=1}^{L} \bigoplus_{k=1}^{2^{l-1}} \mathbf{L}_{l,k}\mathbf{R}_{l,k}^\top \right) \mathbf{x}_{j'} \quad (6)$$

$$s_{\text{BTT}}(\mathbf{x}_j, \mathbf{x}_{j'}) = \mathbf{x}_j^\top \left( \mathbf{P}_L \bigoplus_{k'=1}^{b} \mathbf{L}_{k'} \mathbf{P}_R \bigoplus_{k=1}^{c} \mathbf{R}_k^\top \right) \mathbf{x}_{j'} \quad (7)$$

As shown in Table 1, a BTT matrix with $a = b = c = d = \sqrt{D}$ requires only $O(D^{3/2})$ parameters and FLOPs, but it is full rank. For the MLR version, we set the numbers of blocks to be powers of two: $p_l = 2^{l-1}$. Setting $\sum_l r_l$ to be the "head dimension" $r$, we match the efficiency of standard attention and achieve high or even full rank. In Section 4, we train transformers with these structured scoring functions on in-context regression.

It is also possible to interpret these scoring functions as constructing (higher-dimensional) queries and keys per attention head. See Appendix B for details. Thus our approach is also fully compatible with grouped-query attention (GQA) that shares the same KV transformations across multiple query heads (Ainslie et al., 2023).

### 3.3. MLBTC: Multi-Level Block Tensor Contraction

In this section, we show that both MLR and BTT are special cases of a novel family of structured matrices which we call Multi-Level Block Tensor Contraction (MLBTC). MLBTC naturally encodes either the full rank constraints or the distance-dependent compute bias.

**Definition 3.1.** The Multi-Level Block Tensor Contraction is defined as

$$\text{MLBTC}(\mathbf{L}, \mathbf{R}) = \sum_{l=1}^{L} \alpha_l \mathbf{P}_L \bigoplus_{k'=1}^{p'_l} \mathbf{L}_{l,k'} \mathbf{P}_R \bigoplus_{k=1}^{p_l} \mathbf{R}_{l,k}^\top \quad (8)$$

where $\alpha_l \in \mathbb{R}$, $\mathbf{P}_L$ and $\mathbf{P}_R$ are fixed permutation matrices, $\mathbf{L}_{l,k'} \in \mathbb{R}^{m_{l,k'} \times r'_l}$, $\mathbf{R}_{l,k} \in \mathbb{R}^{n_{l,k} \times r_l}$, $\sum_{k'=1}^{p'_l} m_{l,k'} = m$, $\sum_{k=1}^{p_l} n_{l,k} = n$, and $r'_l p'_l = r_l p_l$. For square matrices, we assume $m = n = D$.

We show in Appendix C that both MLR and BTT are special cases of the MLBTC matrices. Since BTT generalizes

Monarch (Dao et al., 2020), Butterfly (Dao et al., 2020), Kronecker and MLR generalizes Low Rank, MLBTC naturally contains these matrices as special cases.

## 3.4. Encoding Distance-Dependent Compute Bias with MLR Attention

As described in Section 2.3, standard attention lacks a distance-dependent compute bias, and previous work has sought to create one by imposing a sparse structure onto the score matrix $\mathbf{S}$. We impose an MLR structure instead. MLR matrices inherit the advantages of both low rank and block diagonal matrices. Likewise, our construction inherits the advantages of both standard attention (global receptive field) and sliding window attention (distance-dependent compute bias) by organizing the score matrix into hierarchically nested levels.

We first describe our construction in terms of the scoring function. Divide the sequence into blocks. Define $d(j, j')$ to be 2 when tokens $j$ and $j'$ belong to the same block and 1 otherwise. Then we can define a two-level low-rank scoring function as follows:

$$s_{j,j'}(\mathbf{x}_j, \mathbf{x}_{j'}) = \begin{cases} \mathbf{x}_j^\top (\mathbf{L}_1 \mathbf{R}_1^\top) \mathbf{x}_{j'} & d(j, j') = 1 \\ \mathbf{x}_j^\top (\mathbf{L}_1 \mathbf{R}_1^\top + \mathbf{L}_2 \mathbf{R}_2^\top) \mathbf{x}_{j'} & d(j, j') = 2 \end{cases}$$

where $\mathbf{L}_1, \mathbf{R}_1 \in \mathbb{R}^{D \times r_1}$ and $\mathbf{L}_2, \mathbf{R}_2 \in \mathbb{R}^{D \times r_2}$ are low rank factors. The term $\mathbf{L}_2 \mathbf{R}_2^\top$ increases the rank and power of the scoring function, but only for pairs of tokens in the same block. To generalize this construction, we can further divide each block into subblocks. For a pair of tokens in the same subblock, let $d(j, j') = 3$. Define

$$s_{j,j'}(\mathbf{x}_j, \mathbf{x}_{j'}) = \mathbf{x}_j^\top \left( \sum_{l=1}^{d(j,j')} \mathbf{L}_l \mathbf{R}_l^\top \right) \mathbf{x}_{j'} \qquad (9)$$

This definition extends to any number of subdivisions. The expression in Equation (9) is also equivalent to the matrix form presented in Equation (4). See Appendix D.

We can compute the corresponding score matrix efficiently using batched matrix multiplications. As in standard attention, we form query and key matrices $\mathbf{Q} = \mathbf{X}\mathbf{W}_Q$ and $\mathbf{K} = \mathbf{X}\mathbf{W}_K$. We divide $\mathbf{Q}$ into blocks to obtain the left MLR factors and divide $\mathbf{K}$ to obtain the right ones as shown in Figure 1b. Finally, we combine the factors $\mathbf{Q}_{l,k}$ and $\mathbf{K}_{l,k}$ as in the definition of MLR matrices (Equation (4)): $\mathbf{S} = \sum_{l=1}^L \bigoplus_{k=1}^{p_l} \mathbf{Q}_{l,k} \mathbf{K}_{l,k}^\top$ This is equivalent to the score function defined in Equation (9) when $\mathbf{W}_Q = \begin{bmatrix} \mathbf{L}_1 | & \cdots & |\mathbf{L}_L \end{bmatrix}$ and $\mathbf{W}_K = \begin{bmatrix} \mathbf{R}_1 | & \cdots & |\mathbf{R}_L \end{bmatrix}$.

In our construction, the number of levels $L$, the number of blocks in each level $p_1, \ldots, p_L$, and the rank of each level $r_1, \ldots, r_L$ are hyperparameters chosen such that the sequence length is always longer than $\max_l p_l$ and the sum

of the ranks equals the head dimension: $\sum_l r_l = r$. In this paper, we use $L \leq 8$, and we set $p_l = 2^{l-1}$ to create hierarchically nesting levels.[1] We notate the rank allocation as follows: $r_1 | r_2 | \cdots | r_L$. When $L = 1$, MLR attention reduces to standard attention. When $L > 1$, more FLOPs are allocated to computing each local interaction than each global one, which saves compute overall. Standard attention needs $T^2 r$ FLOPs to form $\mathbf{S}$, but MLR attention uses only $T^2 \sum_{l=1}^L \frac{r_l}{2^{l-1}}$.

Another benefit of MLR attention is that it reduces the key cache size during auto-regressive generation. Since $\mathbf{K}_{2,1}$ only pertains to pairs of tokens in the first half of the sequence, the final token will not attend to it; it only needs $\mathbf{K}_{2,2}$ for level 2 attention. For level $l$, we must only retain $\mathbf{K}_{l,p_l}$. Thus, the total key cache size is $\sum_{l=1}^L \text{size}(\mathbf{K}_{l,p_l}) = T \sum_{l=1}^L \frac{r_l}{2^{l-1}}$, which is smaller than the $\text{size}(\mathbf{K}) = O(Tr)$ cache size needed for standard attention. For example, using 8-level MLR with $r_l = r/8$ would yield a 4x savings. Our proposed MLR attention is also compatible with grouped-query attention (Ainslie et al., 2023) if we apply the above transformations to different attention heads, which can lead to further savings in the KV cache size.

In addition, our MLR attention is also compatible with relative positional embedding methods like RoPE (Su et al., 2023). RoPE encodes positional information by rotating query and key vectors by an angle proportional to the token's index in the sequence. Thus RoPE imposes a form of locality bias in the scoring function. In contrast, our MLR attention is not trying to encode relative positional information or boost the attention score between neighboring tokens. Instead, we change the computational cost of the scoring function based on the tokens' positions. Put differently, standard attention (with or without RoPE) uses the same query/key dimension for all pairs of tokens. We effectively use a smaller query/key dimension for tokens that are far apart. This saves FLOPs compared to standard attention, while still allowing high quality attention scores for neighboring tokens.

## 3.5. Practical Considerations

Our attention variants are implemented using batch matrix multiplications. We present the most efficient tensor contraction order in Appendix E. Maximum Update Parameterization ($\mu$P) is a recipe to set the learning rates and initializations of each weight matrix in a neural network for stable feature learning as the width of the network grows (Yang et al., 2022). We describe how to adapt $\mu$P to our structured attention variants, along with our use of normal-

---

[1]It is also possible to select the size of each block in a level dynamically according to the input. For instance, one level could divide the tokens into blocks corresponding to paragraphs, while another could divide it into documents.

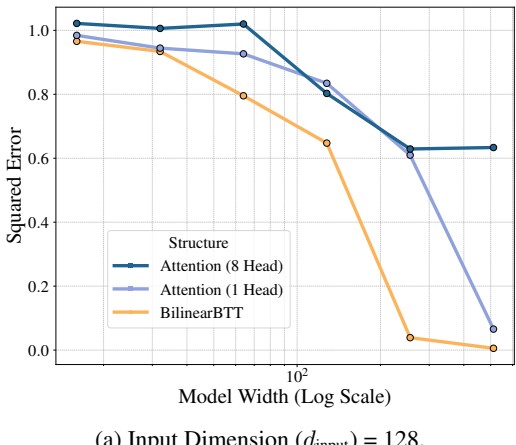

(a) Input Dimension ($d_{\text{input}}$) = 128.

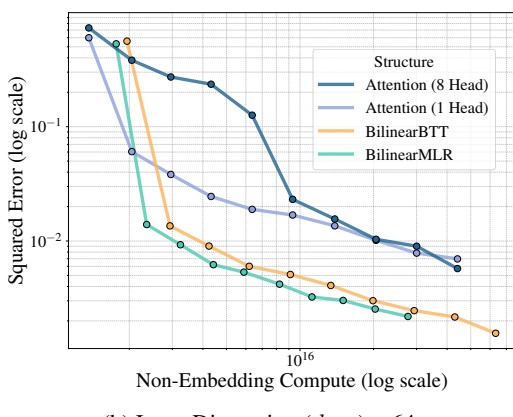

(b) Input Dimension ($d_{\text{input}}$) = 64.

*Figure 3.* **Both Bilinear BTT and Bilinear MLR outperform standard attention for in-context regression tasks**. (a) Bilinear BTT and 1-head attention are full rank, so they can learn the task using a smaller model width (x-axis) than 8-head attention. Here model width refers to the embedding dimension $D$. All points on this figure correspond to models that was trained for the same number of steps. (b) For a fixed model width ($D = 256$), both Bilinear MLR and Bilinear BTT have smaller regression error compared to attention with 1 or 8 heads when controlling for compute. Here our bilinear models use 8 heads.

izization layers, in Appendix F. We also include a figure which shows stable learning rate transfer across width for our MLR attention model in Figure 5 in Appendix G.

## 4. Experiments with the Low Rank Bottleneck

Since both Bilinear BTT and Bilinear MLR in Equations (6) and (7) increase the rank with fewer FLOPs compared to a dense matrix, we test if our new scoring functions based on these structured matrices can resolve rank bottleneck on the in-context regression task based on Garg et al. (2022).

Following Garg et al. (2022), we train transformers on prompts of the form $\mathbf{x}_1, f(\mathbf{x}_1), \mathbf{x}_2, f(\mathbf{x}_2), \dots, \mathbf{x}_N$ to output $f(\mathbf{x}_N)$, where $\mathbf{x}_i \in \mathbb{R}^{d_{\text{input}}}$ and $f(\mathbf{x}_i) = \mathbf{w}^\top \mathbf{x}_i$. The linear functional $\mathbf{w} \sim \mathcal{N}(\mathbf{0}, \mathbf{I}_{d_{\text{input}}})$ is freshly sampled for each prompt, so the model must use in-context learning to infer $\mathbf{w}$ and apply it to $\mathbf{x}_N$. We draw $\mathbf{x}_i \sim \mathcal{N}(\mathbf{0}, \frac{1}{d_{\text{input}}}\mathbf{I}_{d_{\text{input}}})$ and set $N = 2d_{\text{input}}$. We train the model causally on the loss $\frac{1}{N} \cdot \sum_{i=1}^N (\hat{f}(\mathbf{x}_i) - f(\mathbf{x}_i))^2$ where $\hat{f}(\mathbf{x}_i)$ is the model prediction given the first $i - 1$ pairs as context.

Unlike language models, which have token embedding and unembedding layers, we have an input linear layer that maps input points from $\mathbb{R}^{d_{\text{input}}}$ to embeddings in $\mathbb{R}^D$ and an output linear layer that maps embeddings to $\mathbb{R}$. Following $\mu$P (Yang et al., 2022), the last layer is zero-initialized so that $\mathbb{E}[(\hat{f}(\mathbf{x}_i) - f(\mathbf{x}_i))^2] = 1$ at initialization.

In Figure 3, we show that the structured attention variants introduced in Section 3.2 outperform standard attention on this task. In Figure 3a we plot the error of Bilinear BTT attention with 8 heads and standard attention with 1 and 8 heads across a range of model widths $D$. Standard 8-head

attention suffers from a low-rank bottleneck that prevents it from learning this task even with $D = 512$. 1-head attention, for which $r = D$, has a full-rank scoring function. It learns the task at $D = 512$. Bilinear BTT is also full rank, even with 8 heads, so it performs well too. In this case, it needs only $D = 256$. We defer a full set of results across multiple input dimensions $d_{\text{input}}$ to Figure 11 in the appendix.

Figure 3b shows that both BilinearBTT and BilinearMLR attention attain significantly higher accuracy when controlling for training compute. We measure compute in FLOPs, excluding the input and output projection layers (which have the same cost across all these models).[2] We present the full set of sweeps across different model widths $D$ and input dimensions $d_{\text{input}}$ in Figures 7 to 10 in the appendix.

## 5. Experiments with Distance-Dependent Compute Bias

### 5.1. Language Modeling

We train 6-layer transformers with both standard attention and MLR attention on the OpenWebText dataset with a batch size of 4, sequence length $T = 1024$, head dimension $r = 64$, and model width $D \in \{256, 384, 512, 768\}$. The model is trained with AdamW (Loshchilov & Hutter, 2019) and we tune hyperparameters based on $\mu$P (Yang et al., 2022). We use character-level tokenization, which improves

---

[2]We did not optimize the implementations of our methods, so they are somewhat slower in wall-clock time than standard attention. However Figure 12 shows that our methods are still superior controlling for wall-clock time and hardware. Since they use parallelizable tensor operations, we believe better speed is attainable.

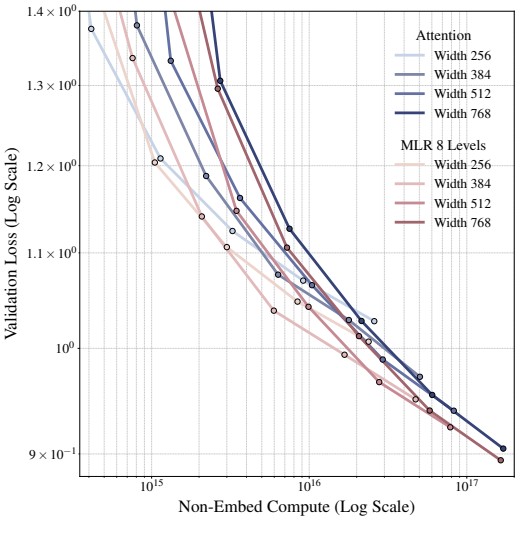

(a) Scaling Law on OpenWebText

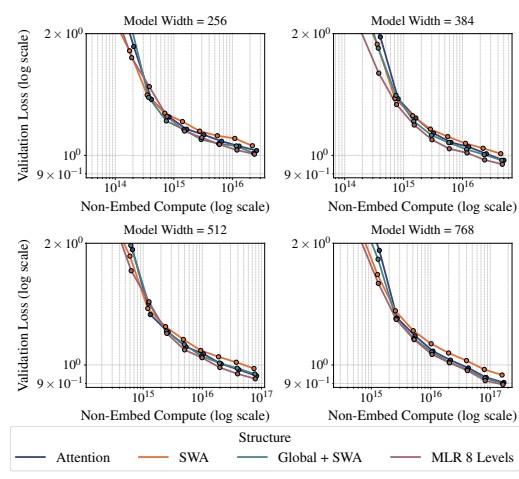

(b) Different Forms of Distance-Dependent Compute

*Figure 4.* **MLR Attention achieves lower validation loss on OpenWebText compared to standard multi-head attention and variants of sliding window attention when controlling for compute.** (a) We plot the validation loss of both MLR attention with 8 levels and standard attention against compute. We vary over 4 values of model width $D \in \{256, 384, 512, 768\}$. MLR attention outperforms standard attention across model widths. (b) We compare MLR attention with variants of sliding window attention (SWA) that also encodes distance-dependent compute bias. Given the training sequence length $T = 1024$ and a 6-layers transformer model, SWA refers to 6 layers of sliding window attention with window size $T' = 128$. Global + SWA refers to a combination of standard attention and sliding window attention with $T' = 128$. The first and fourth layers of the model are standard attention and the rest are sliding window attention. We observe that across the model width $D$, MLR attention outperforms standard attention, SWA, and Global + SWA when controlling for compute.

our ability to study scaling with model size with limited compute budget (Potapczynski et al., 2024).

In Figure 4a, we compare the language modeling performance of standard attention to 8-level MLR attention across model widths $D$. The rank allocation is 32|8|6|4|4|4|4|2, for a total of $r = 64$. MLR attention outperforms standard attention across all model widths throughout training, and thus exhibits a better scaling law.

In Figure 4b, we compare MLR attention to standard (global) attention, sliding window attention (SWA) (Child et al., 2019; Beltagy et al., 2020), and an architecture with alternating layers of global and sliding window attention. Like MLR, these architectures aim to take advantage of locality. Across model widths and training time, MLR attention performs best when controlling for compute.

**5.2. Time Series Forecasting**

We also compare standard attention and MLR attention on time series prediction. The Electricity Transformer Temperature (ETT) dataset (Zhou et al., 2021) tracks fluctuations of oil temperature along with six additional power load features across time. As shown in Figure 14, MLR attention gradually outperforms standard attention in oil temperature prediction accuracy as the time horizon grows. We defer additional details to Appendix J.

## 6. Related Work

Previous work has replaced dense neural network weight matrices by structured matrices—including low displacement rank (Thomas et al., 2018), low-rank plus sparse or diagonal (Han et al., 2024; Wei et al., 2024), Monarch (Dao et al., 2022), BTT (Qiu et al., 2024), and others (Potapczynski et al., 2024)—for greater efficiency, as well as for fine-tuning (Hu et al., 2021; Sehanobish et al., 2024) and compression-based generalization bounds (Lotfi et al., 2024a;b).

Structured matrices have also been used in place of the attention matrix. Chen et al. (2021) approximates the attention matrix by a sparse plus low-rank matrix. Hwang et al. (2024) proposes the matrix mixer framework that identifies sequence mixers like Attention, S4 (Gu et al., 2022), H3 (Fu et al., 2023), Hyena (Poli et al., 2023), and Mamba (Gu & Dao, 2024; Dao & Gu, 2024) with particular structured families. As in our work, they show that structured matrices offer a systematic and principled way to explore the design space.

Following sliding window attention (Child et al., 2019; Beltagy et al., 2020), many methods have been proposed to

combine local attention with some global component (Beltagy et al., 2020; Hatamizadeh et al., 2023; Arora et al., 2024; Behrouz et al., 2024a; Huang et al., 2023, §3.1). Like our MLR attention, other work has constructed hierarchically nested levels of progressively longer-range attention (Ye et al., 2019; Zhu & Soricut, 2021; Ren et al., 2021; Huang et al., 2023, §3.2). Unlike us, their goal is to make attention sublinear in the sequence length. Thus, they must aggregate tokens in each block together (e.g. using max-pool), departing significantly from the form of standard attention. Our study of the low rank bottleneck in attention follows that of Bhojanapalli et al. (2020); Sanford et al. (2023) and Amsel et al. (2024), though past work has not tried modifying attention to fix it. In addition, many works have tried to reduce the quadratic dependence on the sequence length of standard attention by approximating the softmax computations with random features (Choromanski et al., 2022), removing softmax (Katharopoulos et al., 2020), or in general reducing KV caches in the model architecture (Ainslie et al., 2023; Yuan et al., 2025).

## 7. Discussion

In this paper, we use structured matrices to design variants of attention with inductive biases that are beneficial for certain tasks. Bilinear BTT and Bilinear MLR attention resolve the low-rank bottleneck for in-context regression. Both bilinear structures allow the scoring function to be more expressive compared to low rank matrices while being efficient. While some tasks including language modeling do not seem to suffer from the low rank bottleneck, we believe our technique can improve transformers' performance on a variety of datasets that are intrinsically high-dimensional, especially scientific and PDE data. In addition, we use MLR attention to create a flexible distance-dependent compute bias that improves performance on language modeling and time series prediction with less FLOPs. Here too, we believe our methods can have further impact on point cloud data and tasks like code generation, which has not only a strong positional structure, but multiple levels of nested structures such as functions, classes, files, and packages. Furthermore, we define the Multi-Level Block Tensor Contraction (MLBTC) class in Section 3.3 to generalize BTT and MLR. We hope future work will explore the benefits and inductive biases of MLBTC's greater flexibility in attention and elsewhere.

In this work, we study the attention score matrix $\mathbf{X}\mathbf{W}_{Q_i}\mathbf{W}_{K_i}^\top\mathbf{X}^\top$ and the low-rank scoring function at its center. Similar techniques could be applied to the other part of each attention head, defined by the low-rank matrix $\mathbf{W}_{V_i}\mathbf{W}_{O_i}^\top$. Rather than customizing each head in a transformer the same way, different structures could be used for different heads, as in Xu et al. (2025), to promote a better division of labor. Finally, our techniques could be adopted for

fine-tuning or model compression rather than pre-training. We defer further exploration of these aspects to future work.

## Acknowledgements

We thank Alan Amin and Hoang Phan for helpful discussions and anonymous reviewers for helpful feedback. This work was supported in part by NSF CAREER IIS-2145492, NSF CDS&E-MSS 2134216, NSF HDR-2118310, NSF Award 1922658, BigHat Biosciences, Capital One, and an Amazon Research Award.

## Impact Statement

This paper presents work whose goal is to advance the field of Machine Learning. There are many potential societal consequences of our work, none which we feel must be specifically highlighted here.

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

# A. Rectangular Structured Matrices

In this section, we discuss the extension of Multi-Level Low Rank (MLR) matrix and Block Tensor Train (BTT) matrix to rectangular shape $m \times n$. The remaining appendix sections also follow the notations defined for rectangular MLR and rectangular BTT.

## A.1. Rectangular MLR

A MLR matrix with shape $m \times n$ is given by

$$\sum_{l=1}^{L} \bigoplus_{k=1}^{p_l} \mathbf{L}_{l,k} \mathbf{R}_{l,k}^{\top} \tag{10}$$

where $\bigoplus_{k=1}^{p_l} \mathbf{L}_{l,k} \mathbf{R}_{l,k}^{\top} \in \mathbb{R}^{m \times n}$ is a block diagonal matrix with $p_l$ blocks at level $l$. Each block $\mathbf{L}_{l,k} \mathbf{R}_{l,k}^{\top}$ is a low rank product for $\mathbf{L}_{l,k} \in \mathbb{R}^{m_{l,k} \times r_l}$ and $\mathbf{R}_{l,k} \in \mathbb{R}^{n_{l,k} \times r_l}$. By construction, we have $\sum_{k=1}^{p_l} m_{l,k} = m$ and $\sum_{k=1}^{p_l} n_{l,k} = n$. The MLR rank of the matrix $\mathbf{W}$ is defined as $r_{\text{MLR}} = \sum_{l=1}^{L} r_l$.

## A.2. Rectangular BTT

A BTT matrix of shape $m \times n$ is given by

$$\mathbf{P}_L \bigoplus_{k'=1}^{b} \mathbf{L}_{k'} \mathbf{P}_R \bigoplus_{k=1}^{c} \mathbf{R}_k^{\top} \tag{11}$$

where $\mathbf{R}_k \in \mathbb{R}^{d \times bs}, \mathbf{L}_{k'} \in \mathbb{R}^{a \times cs}$, $ab = m$, $cd = n$. The extra dimension $s$ is the BTT rank. $\mathbf{P}_L$ permutes the rows by rearranging dimension $ba$ into $ab$. This permutation is equivalent to reshaping a vector $\mathbf{z} \in \mathbb{R}^{ba}$ into a matrix $\mathbf{Z} \in \mathbb{R}^{b \times a}$, tranposing to get $\mathbf{Z}^{\top} \in \mathbb{R}^{a \times b}$, and reshaping it back to a vector $\mathbf{z}' \in \mathbb{R}^{ab}$. $\mathbf{P}_R$ permutes the rows by rearranging dimension $cbs$ into $bcs$.

# B. Bilinear Attention Scoring Functions with Structured Matrices

In this section, we describe how the bilinear BTT and bilinear MLR attention scoring functions can be interpreted in terms of queries and keys with projection matrices that are structured instead of dense.

## B.1. Notations

The appendix section adopts the following notation. Let $B$ be the batch size, $T$ be the sequence length during training, $\mathbf{W}_Q, \mathbf{W}_K, \mathbf{W}_V \in \mathbb{R}^{D \times r}$ be projection matrices, and $H$ be the number of heads such that $D = Hr$. For any $\mathbf{x}_i \in \mathbb{R}^{D \times 1}$, we obtain $\mathbf{q}_i, \mathbf{k}_i, \mathbf{v}_i \in \mathbb{R}^{r \times 1}$ by the projections $\mathbf{q}_i = \mathbf{W}_Q^{\top} \mathbf{x}_i, \mathbf{k}_i = \mathbf{W}_K^{\top} \mathbf{x}_i, \mathbf{v}_i = \mathbf{W}_V^{\top} \mathbf{x}_i$. Since matrices in attention usually have square shapes, we assume $m = n = D$.

## B.2. Bilinear MLR

Let $\mathbf{X} \mathbf{W}_Q \mathbf{W}_K^{\top} \mathbf{X}^{\top}$ be the form of attention matrix per attention head. We can replace the low rank structure $\mathbf{W}_Q \mathbf{W}_K^{\top}$ with $\text{MLR}(\mathbf{W}_Q, \mathbf{W}_K)$ and obtain the Bilinear MLR projections:

$$\mathbf{Q}_l^{\text{MLR}} = \mathbf{P}_1 \left( \mathbf{X} \bigoplus_{k=1}^{p_l} \mathbf{W}_{Q_{l,k}} \right) \tag{12}$$

$$\mathbf{K}_l^{\text{MLR}} = \mathbf{P}_2 \left( \bigoplus_{k=1}^{p_l} \mathbf{W}_{K_{l,k}}^{\top} \mathbf{X}^{\top} \right) \tag{13}$$

where $\mathbf{W}_{Q_{l,k}} \in \mathbb{R}^{m_{l,k} \times Hr_l}, \mathbf{W}_{K_{l,k}} \in \mathbb{R}^{n_{l,k} \times Hr_l}$, and thus $\bigoplus_{k=1}^{p_l} \mathbf{W}_{Q_{l,k}} \in \mathbb{R}^{m \times p_l Hr_l}, \bigoplus_{k=1}^{p_l} \mathbf{W}_{K_{l,k}} \in \mathbb{R}^{n \times p_l Hr_l}$. Thus the shape of the product $\mathbf{X} \bigoplus_{k=1}^{p_l} \mathbf{W}_{Q_{l,k}}$ is $(BT, p_l Hr_l)$ and the shape of the product $\bigoplus_{k=1}^{p_l} \mathbf{W}_{K_{l,k}}^{\top} \mathbf{X}^{\top}$ is $(p_l Hr_l, BT)$. The permutation matrix $\mathbf{P}_1$ rearranges the shape $(BT, p_l Hr_l)$ into $(B, H, T, r_l p_l)$. $\mathbf{P}_2$ rearranges the shape $(p_l Hr_l, BT)$ into $(B, H, r_l p_l, T)$.

The final output of the BilinearMLR sequence mixing operation $\mathbf{O} \in \mathbb{R}^{B \times H \times T \times T}$ is given by

$$\mathbf{O}_{\alpha\beta} = \sigma\left(\mathbf{M} \circ \sum_{l=1}^{L} \mathbf{Q}_{l\alpha\beta}^{\text{MLR}} \mathbf{K}_{l\alpha\beta}^{\text{MLR}}\right) \tag{14}$$

where $\mathbf{Q}_{l\alpha\beta}^{\text{MLR}} \in \mathbb{R}^{T \times r_l p_l}$ and $\mathbf{K}_{l\alpha\beta}^{\text{MLR}} \in \mathbb{R}^{r_l p_l \times T}$. In practice we assume $p_l = 2^{l-1}$, and $m_{l,k} = n_{l,k} = D/2^{l-1}$.

### B.3. Bilinear BTT

We can replace the low rank structure $\mathbf{W}_Q \mathbf{W}_K^\top$ with $\text{BTT}(\mathbf{W}_Q, \mathbf{W}_K)$ and obtain the Bilinear BTT projections:

$$\mathbf{Q}^{\text{BTT}} = \mathbf{P}_3\left(\mathbf{X} \bigoplus_{k=1}^{b} \mathbf{W}_{Q_k}\right) \tag{15}$$

$$\mathbf{K}^{\text{BTT}} = \mathbf{P}_4\left(\bigoplus_{k=1}^{c} \mathbf{W}_{K_k}^\top \mathbf{X}^\top\right) \tag{16}$$

where we assume $\mathbf{X} \in \mathbb{R}^{BT \times D}$, $\mathbf{W}_{Q_k} \in \mathbb{R}^{a \times Hcs}$, $\mathbf{W}_{K_k} \in \mathbb{R}^{d \times Hbs}$ and thus $\bigoplus_{k=1}^{b} \mathbf{W}_{Q_k} \in \mathbb{R}^{ba \times bHcs}$, $\bigoplus_{k=1}^{c} \mathbf{W}_{K_k}^\top \in \mathbb{R}^{cHbs \times cd}$. Here we assume $D = ab = cd$. Thus the shape of the product $\mathbf{X} \bigoplus_{k=1}^{b} \mathbf{W}_{Q_k}$ is $(BT, bHcs)$ and the shape of the product $\bigoplus_{k=1}^{c} \mathbf{W}_{K_k}^\top \mathbf{X}^\top$ is $(cHbs, BT)$. The permutation matrix $\mathbf{P}_3$ rearrange the shape $(BT, bHcs)$ into $(B, H, T, sbc)$. $\mathbf{P}_4$ rearrange the shape $(cHbs, BT)$ into $(B, H, sbc, T)$.

Let $\alpha$ denotes the batch index and $\beta$ denotes the head index such that $\mathbf{Q}_{\alpha\beta}^{\text{BTT}} \in \mathbb{R}^{T \times sbc}$ and $\mathbf{K}_{\alpha\beta}^{\text{BTT}} \in \mathbb{R}^{sbc \times T}$. The final output of the bilinear BTT $\mathbf{O} \in \mathbb{R}^{B \times H \times T \times T}$ is compute as

$$\mathbf{O}_{\alpha\beta} = \sigma(\mathbf{M} \circ \mathbf{Q}_{\alpha\beta}^{\text{BTT}} \mathbf{K}_{\alpha\beta}^{\text{BTT}}) \tag{17}$$

where $\sigma$ is the softmax function and $\mathbf{M}$ is the lower diagonal causal mask matrix. Both $B$ and $H$ are extra batch dimensions that can be computed efficiently using the batch matrix multiplication primitives.

In practice, it's always true that $sbc \gg r$ where $r$ is the head dimension for standard multi-head attention.

## C. Multi-Level Block Tensor Contraction

To see that MLBTC captures all MLR matrices (including low-rank matrices as a special case), set $\alpha_l = 1$, $\mathbf{P}_L = \mathbf{P}_R = \mathbf{I}$, $k' = k$, $p'_l = p_l$, and $r'_l = r_l$ in Equation (8). To see that it captures all BTT matrices (including Monarch, Butterfly, and Kronecker), set $\alpha_l = 0$ for all but one $l$ and select the dimensions to match Equation (5): $p'_l = b$, $p_l = c$, $r'_l = cs$, $r_l = bs$, $n_{l,k} = d$, and $m_{l,k'} = a$.

Now consider square matrices such that $m = n = D$. Since BTT can approximate arbitrary dense matrices of shape $D \times D$ based on Qiu et al. (2024), MLBTC can also express any $D \times D$ matrices with large enough $r_l$ and $r'_l$. Like MLR and BTT, MLBTC matrices can be used to define an efficient full-rank scoring function. We leave this generalization to future work.

## D. MLR Attention Derivation

In this section, we present Lemma D.1 that connects the matrix form of MLR and its scoring function formula.

**Lemma D.1.** *The $(j, j')$ entry of the scoring matrix $\mathbf{S} = \sum_{l=1}^{L} \bigoplus_{k=1}^{p_l} \mathbf{Q}_{l,k} \mathbf{K}_{l,k}^\top$ has the form $\mathbf{x}_j^\top \left(\sum_{l=1}^{d(j,j')} \mathbf{L}_l \mathbf{R}_l^\top\right) \mathbf{x}_{j'}$ as shown in the right hand side of Equation (9).*

*Proof.* For simplicity, assume there are just two levels. So Equation (9) reduces to the following expression:

$$s_{j,j'}(\mathbf{x}_j, \mathbf{x}_{j'}) = \mathbf{x}_j^\top (\mathbf{L}_1 \mathbf{R}_1^\top + \mathbf{L}_2 \mathbf{R}_2^\top) \mathbf{x}_{j'} \tag{18}$$

Divide $\mathbf{X} = \begin{bmatrix} \mathbf{X}_1 \\ \mathbf{X}_2 \end{bmatrix}$ into blocks. $\mathbf{X}_1$ is the first half of the sequence and $\mathbf{X}_2$ corresponds to the second half. Divide

$\mathbf{W}_Q = \begin{bmatrix} \mathbf{L}_1 & \mathbf{L}_2 \end{bmatrix}$. Thus

$$\mathbf{Q} = \mathbf{X}\mathbf{W}_Q = \begin{bmatrix} \mathbf{X}_1\mathbf{L}_1 & \mathbf{X}_1\mathbf{L}_2 \\ \mathbf{X}_2\mathbf{L}_1 & \mathbf{X}_2\mathbf{L}_2 \end{bmatrix} \tag{19}$$

Now divide $\mathbf{Q}$ into 3 named blocks according to Figure 1b:

$$\mathbf{Q}_{11} = \mathbf{X}\mathbf{L}_1 \tag{20}$$
$$\mathbf{Q}_{21} = \mathbf{X}_1\mathbf{L}_2 \tag{21}$$
$$\mathbf{Q}_{22} = \mathbf{X}_2\mathbf{L}_2 \tag{22}$$

Analogously

$$\mathbf{K}_{11} = \mathbf{X}\mathbf{R}_1 \tag{23}$$
$$\mathbf{K}_{21} = \mathbf{X}_1\mathbf{R}_2 \tag{24}$$
$$\mathbf{K}_{22} = \mathbf{X}_2\mathbf{R}_2 \tag{25}$$

Finally, plug the above into the definition of $\mathbf{S}$:

$$\mathbf{S} = \mathbf{Q}_{11}\mathbf{K}_{11}^\top + \begin{bmatrix} \mathbf{Q}_{21}\mathbf{K}_{21}^\top & \mathbf{0} \\ \mathbf{0} & \mathbf{Q}_{22}\mathbf{K}_{22}^\top \end{bmatrix} = \mathbf{X}\mathbf{L}_1\mathbf{R}_1^\top\mathbf{X}^\top + \begin{bmatrix} \mathbf{X}_1\mathbf{L}_2\mathbf{R}_2^\top\mathbf{X}_1^\top & \mathbf{0} \\ \mathbf{0} & \mathbf{X}_2\mathbf{L}_2\mathbf{R}_2^\top\mathbf{X}_2^\top \end{bmatrix} \tag{26}$$

Consider the $(j, j')$ entry of $\mathbf{S}$. If they're in different blocks, then it's $\mathbf{x}_j^\top \mathbf{L}_1 \mathbf{R}_1^\top \mathbf{x}_j$. If they're in the same block, it's $\mathbf{x}_j^\top \mathbf{L}_1 \mathbf{R}_1^\top \mathbf{x}_j + \mathbf{x}_j^\top \mathbf{L}_2 \mathbf{R}_2^\top \mathbf{x}_j$. The same reasoning applies when we have more than two levels. $\square$

## E. Optimal Tensor Contraction Order

### E.1. Bilinear MLR

In this section, we compute the costs to do Bilinear MLR per attention head assuming batch size equal to $1$. We write bilinear MLR as

$$\mathbf{X}\text{MLR}(\mathbf{W}_Q, \mathbf{W}_K)\mathbf{X}^\top = \mathbf{X}\Big(\sum_{l=1}^L \bigoplus_{k=1}^{p_l} \mathbf{W}_{Q_{l,k}}\mathbf{W}_{K_{l,k}}^\top\Big)\mathbf{X}^\top \tag{27}$$

$$= \sum_{l=1}^L \mathbf{X}\Big(\bigoplus_{k=1}^{p_l} \mathbf{W}_{Q_{l,k}}\mathbf{W}_{K_{l,k}}^\top\Big)\mathbf{X}^\top \tag{28}$$

$$= \sum_{l=1}^L \Big(\mathbf{X}\bigoplus_{k=1}^{p_l} \mathbf{W}_{Q_{l,k}}\Big)\Big(\bigoplus_{k=1}^{p_l} \mathbf{W}_{K_{l,k}}^\top\mathbf{X}^\top\Big) \tag{29}$$

$$= \sum_{l=1}^L \mathbf{X}\Big(\bigoplus_{k=1}^{p_l} \mathbf{W}_{Q_{l,k}} \bigoplus_{k=1}^{p_l} \mathbf{W}_{K_{l,k}}^\top\mathbf{X}^\top\Big) \tag{30}$$

$$= \mathbf{X}\sum_{l=1}^L \Big(\bigoplus_{k=1}^{p_l} \mathbf{W}_{Q_{l,k}} \bigoplus_{k=1}^{p_l} \mathbf{W}_{K_{l,k}}^\top\mathbf{X}^\top\Big) \tag{31}$$

We summarize the FLOPs for each of these tensor contraction orders in Table 2. We find that

$$\sum_{l=1}^L \Big(\mathbf{X}\bigoplus_{k=1}^{p_l} \mathbf{W}_{Q_{l,k}}\Big)\Big(\bigoplus_{k=1}^{p_l} \mathbf{W}_{K_{l,k}}^\top\mathbf{X}^\top\Big) \tag{32}$$

has minimal compute costs of $2TDr + T^2\sum_{l=1}^L 2^{l-1}r_l$. This is the form of bilinear MLR we used in all of our experiments.

*Table 2.* FLOPs for Bilinear MLR with Different Tensor Contraction Ordering

| TENSOR CONTRACTION ORDER | FLOPs |
| --- | --- |
| (LOW RANK) $\mathbf{X}\mathbf{W}_Q\mathbf{W}_K^\top\mathbf{X}^\top$ | $T^2 r + 2TDr$ |
| $\mathbf{X}\left(\sum_{l=1}^L \bigoplus_{k=1}^{p_l} \mathbf{W}_{Q_{l,k}}\mathbf{W}_{K_{l,k}}^\top\right)\mathbf{X}^\top$ | $T^2 D + TD^2 +$ $D^2(\sum_{l=1}^L \frac{r_l}{2^{l-1}} + \sum_{l=1}^L \frac{1}{2^l} - \frac{1}{2})$ |
| $\sum_{l=1}^L \mathbf{X}\left(\bigoplus_{k=1}^{p_l} \mathbf{W}_{Q_{l,k}}\mathbf{W}_{K_{l,k}}^\top\right)\mathbf{X}^\top$ | $T^2 LD + \sum_{l=1}^L \frac{D^2 r_l}{2^{l-1}} + \frac{TD^2}{2^{l-1}}$ |
| $\sum_{l=1}^L \left(\mathbf{X}\bigoplus_{k=1}^{p_l} \mathbf{W}_{Q_{l,k}}\right)\left(\bigoplus_{k=1}^{p_l} \mathbf{W}_{K_{l,k}}^\top\mathbf{X}^\top\right)$ | $2TDr + T^2 \sum_{l=1}^L 2^{l-1}r_l$ |
| $\sum_{l=1}^L \mathbf{X}\left(\bigoplus_{k=1}^{p_l} \mathbf{W}_{Q_{l,k}} \bigoplus_{k=1}^{p_l} \mathbf{W}_{K_{l,k}}^\top\mathbf{X}^\top\right)$ | $LT^2 D + 2TDr$ |
| $\mathbf{X}\sum_{l=1}^L \left(\bigoplus_{k=1}^{p_l} \mathbf{W}_{Q_{l,k}} \bigoplus_{k=1}^{p_l} \mathbf{W}_{K_{l,k}}^\top\mathbf{X}^\top\right)$ | $T^2 D + 2TDr$ |

### E.2. Bilinear BTT

In this section, we analyze the FLOPs required for bilinear BTT assuming batch size equal to 1 and 1 attention head. We write bilinear BTT as

$$\mathbf{X}\text{BTT}(\mathbf{W}_Q, \mathbf{W}_K)\mathbf{X}^\top = \left(\mathbf{X}\mathbf{P}_L \bigoplus_{k=1}^b \mathbf{W}_{Q_k}\right)\left(\mathbf{P}_R \bigoplus_{k=1}^c \mathbf{W}_{K_k}^\top\mathbf{X}^\top\right) \tag{33}$$

$$= \mathbf{X}\left(\mathbf{P}_L \bigoplus_{k=1}^b \mathbf{W}_{Q_k}\mathbf{P}_R \bigoplus_{k=1}^c \mathbf{W}_{K_k}^\top\mathbf{X}^\top\right) \tag{34}$$

where $\mathbf{X} \in \mathbb{R}^{T\times D}$, $\mathbf{W}_{Q_k} \in \mathbb{R}^{a\times cs}$, $\mathbf{W}_{K_k} \in \mathbb{R}^{d\times bs}$ with $D = ab = cd$. In Table 3, we list out the FLOP counts with the simplifying assumption of $a = b = c = d = \sqrt{D}$. Thus we should be using the form

$$\mathbf{X}\left(\mathbf{P}_L \bigoplus_{k=1}^b \mathbf{W}_{Q_k}\mathbf{P}_R \bigoplus_{k=1}^c \mathbf{W}_{K_k}^\top\mathbf{X}^\top\right) \tag{35}$$

with FLOPs $T^2 D + 2sTD^{3/2}$.

*Table 3.* FLOPs for Bilinear BTT with Different Tensor Contraction Ordering

| TENSOR CONTRACTION ORDER | FLOPs |
| --- | --- |
| $\left(\mathbf{X}\mathbf{P}_L \bigoplus_{k=1}^b \mathbf{W}_{Q_k}\right)\left(\mathbf{P}_R \bigoplus_{k=1}^c \mathbf{W}_{K_k}^\top\mathbf{X}^\top\right)$ | $sT^2 D + 2sTD^{3/2}$ |
| $\mathbf{X}\left(\mathbf{P}_L \bigoplus_{k=1}^b \mathbf{W}_{Q_k}\mathbf{P}_R \bigoplus_{k=1}^c \mathbf{W}_{K_k}^\top\mathbf{X}^\top\right)$ | $T^2 D + 2sTD^{3/2}$ |

## F. Practical Considerations.

### F.1. Maximal Update Parameterization ($\mu$P)

For a dense matrix $\mathbf{W} \in \mathbb{R}^{d_\text{out}\times d_\text{in}}$ and an input vector $\mathbf{x} \in \mathbb{R}^{d_\text{in}}$, the output hidden state $\mathbf{h} \in \mathbb{R}^{d_\text{out}}$ is computed as $\mathbf{h} = \mathbf{W}\mathbf{x}$. $\mu$P requires both $\mathbf{h}$ and the gradient update $\Delta\mathbf{h}$ to have norm $\|\mathbf{h}\|_2 = \|\Delta\mathbf{h}\|_2 = \Theta(\sqrt{d_\text{out}})$ for stable feature learning under width scaling. This is equivalent to imposing the spectral norm constraints on the weight matrix such that $\|\mathbf{W}\|_* = \|\Delta\mathbf{W}\|_* = \Theta(\sqrt{d_\text{out}/d_\text{in}})$ (Yang et al., 2024). Thus the weight matrix $\mathbf{W}$ should be initialized from $\mathcal{N}(0, \sigma_\mathbf{W}^2)$ for $\sigma_\mathbf{W} = \Theta(1/\sqrt{d_\text{in}} \cdot \min\{1, \sqrt{d_\text{out}/d_\text{in}}\})$ with learning rate $\eta_\mathbf{W} = \Theta(d_\text{out}/d_\text{in})$. For AdamW (Loshchilov & Hutter, 2019), the learning rate scales as $\eta_\mathbf{W} = \Theta(1/d_\text{in})$ (Yang et al., 2022). Qiu et al. (2024) develops $\mu$P for structured linear layers that

are expressible as compositions of dense batched matrix multiplications and reshapes. Since our structured attention variants fit this form, they are amenable to $\mu$P.

For bilinear MLR with the optimal tensor contraction order shown in Equation (32), we apply $\mu$P on the dense matrices $\mathbf{W}_{Q_{l,k}} \in \mathbb{R}^{m_{l,k} \times Hr_l}$ and $\mathbf{W}_{K_{l,k}} \in \mathbb{R}^{n_{l,k} \times Hr_l}$ where we assume $m_{l,k} = n_{l,k} = D/p_l$. Since the fan-in dimension for both $\mathbf{W}_{Q_{l,k}}$ and $\mathbf{W}_{K_{l,k}}$ are $D/p_l$, we initialize them as $\mathbf{W}_{Q_{l,k}}, \mathbf{W}_{K_{l,k}} \sim \mathcal{N}(\mathbf{0}, \frac{p_l}{D}\mathbf{I})$. For learning rate, we first select a base value $\eta_{\text{base}} \in \{0.001, 0.0005, 0.0001, 0.00005, 0.00001\}$ at a certain model width $D_1$. Here the model width is the embedding dimension of the transformer model. Then for any model width $D_2 \gg D_1$, we set the learning rate $\eta_{\mathbf{W}_{Q_{l,k}}}$ and $\eta_{\mathbf{W}_{K_{l,k}}}$ to be

$$\eta_{\mathbf{W}_{Q_{l,k}}} = \eta_{\mathbf{W}_{K_{l,k}}} = \eta_{\text{base}} \frac{D_1}{D_2/p_l} \tag{36}$$

for optimal learning rate transfer across width.

Bilinear BTT follows the optimal tensor contraction order in Equation (35) where $\mathbf{W}_{Q_k} \in \mathbb{R}^{a \times cs}$ and $\mathbf{W}_{K_k} \in \mathbb{R}^{d \times bs}$. Thus we initialize them as $\mathbf{W}_{Q_k} \sim \mathcal{N}(\mathbf{0}, \frac{1}{cs}\mathbf{I})$ and $\mathbf{W}_{K_k} \sim \mathcal{N}(\mathbf{0}, \frac{1}{d}\mathbf{I})$. The optimal learning rates are given by

$$\eta_{\mathbf{W}_{Q_k}} = \eta_{\text{base}} \frac{D_1}{cs} \tag{37}$$

$$\eta_{\mathbf{W}_{K_k}} = \eta_{\text{base}} \frac{D_1}{a} \tag{38}$$

Additionally, we apply RMS normalizations on the weight matrices $\mathbf{W}_{Q_k}$ and $\mathbf{W}_{K_k}$ for training stability as shown in Qiu et al. (2024). The rest of the parameters in the transformer models follow the recipe in Yang et al. (2022). For MLR attention, we note that the fan-in dimensions are unchanged and thus it follows the $\mu$P recipe for a standard transformer architecture.

### F.2. QK LayerNorm

Let $\text{LN}(\cdot)$ denote layer normalization. We apply normalizations to bilinear MLR over our optimal tensor contraction order to get

$$\sum_{l=1}^{L} \text{LN}\left(\mathbf{X} \bigoplus_{k=1}^{p_l} \mathbf{W}_{Q_{l,k}}\right) \text{LN}\left(\bigoplus_{k=1}^{p_l} \mathbf{W}_{K_{l,k}}^{\top} \mathbf{X}^{\top}\right) * \frac{C}{r_l p_l} \tag{39}$$

for some tunable constant $C$. We also apply layer normalization to bilinear BTT and get

$$\text{LN}\left(\mathbf{X}\right) \text{LN}\left(\mathbf{P}_L \bigoplus_{k=1}^{b} \mathbf{W}_{Q_k} \mathbf{P}_R \bigoplus_{k=1}^{c} \mathbf{W}_{K_k}^{\top} \mathbf{X}^{\top}\right) * \frac{C^*}{ab} \tag{40}$$

for some tunable constant $C^*$. In standard attention, the score matrix is normalized by $1/\sqrt{r}$ before applying softmax. $\mu$P recommends normalizing by $1/r$ instead (Yang et al., 2022). We incorporate the $\mu$P version of normalization into our architectures. That is, for Bilinear MLR and Bilinear BTT, we use $C/r_l p_l$ and $C^*/ab$ respectively instead of $C/\sqrt{r_l p_l}$ and $C^*/\sqrt{ab}$.

## G. $\mu$P Additional Figures

In this section we show that under the maximum update parametrization ($\mu$P), both our MLR attention and standard attention has stable learning rate across width as shown in Figure 5.

## H. ICL Additional Figures

In this section we provide additional figures of in-context regression performance as a function of compute and training steps across different input dimensions, model widths, and types of scoring functions shown in Figures 7 to 11. We also provide error bars for Figure 2 as shown in Figure 6. Due to compute budget constraints, we only plotted the error bars when the input dimension is 16.

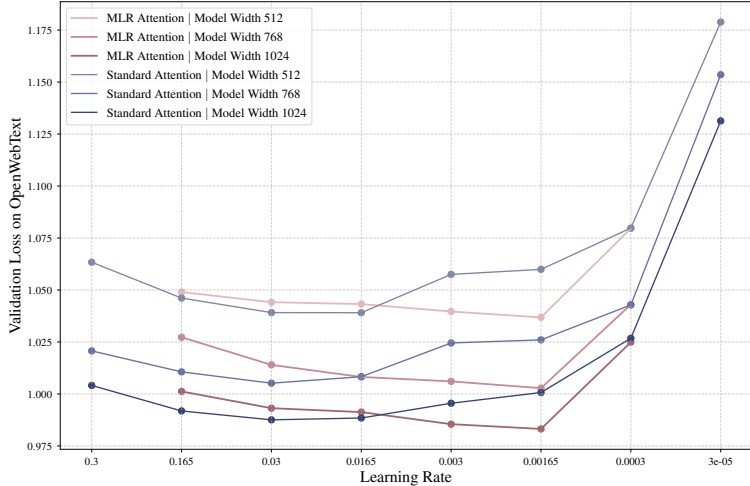

*Figure 5.* **We show the validation loss of an 8-Level MLR attention and standard attention on OpenWebText across a variety of learning rates.** Here we sweep over model widths $D \in \{512, 768, 1024\}$ with a reduced context length of 256 due to compute constraints. As the figure shows, our MLR attention shares the same optimal learning rate across model width, and it's also consistently better than standard attention when both are properly tuned.

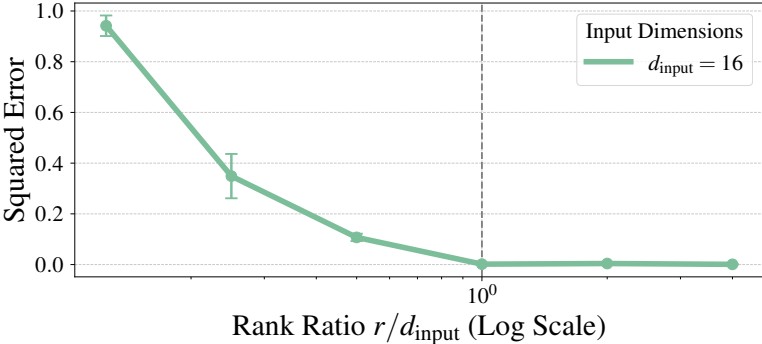

*Figure 6.* **The performance of standard multi-head attention as a function of the rank ratio.** We take the runs from Figure 2 and plot error bars when the input dimension is 16, i.e. $d_{\text{input}} = 16$.

## I. Language Modeling Additional Figures

In this section we present additional figures for language modeling performances between MLR attention and variants of sliding window attention across model widths shown in Figure 13.

## J. Time Series Forecasting Additional Figures

We train a transformer model with 2 encoder layers, embedding dimension $D = 512$, and 8 attention heads on the ETTh1 subset of the ETT dataset. The ETTh1 dataset records the oil temperature and six power load features collected each hour from different electric power transformer stations (Zhou et al., 2021). The dataset contains records with a variety of time horizons. We select horizons of $T \in \{96, 192, 336\}$ hours. Thus, the time series given to the transformer models has sequence length $T$ with feature dimension 7.

The transformer model we used has an input linear projection layer that maps points in $\mathbb{R}^7$ to $\mathbb{R}^D$ and an output linear projection layer that maps points from $\mathbb{R}^D$ to $\mathbb{R}^7$. Following Olivares et al. (2022), we use Ray Tune (Liaw et al., 2018) to sweep over optimal learning rates in the range of $\{0.00001, 0.0002, 0.005\}$ and SGD iterations in $\{200, 1000\}$. In Figure 14, we plot the relative improvement of MLR attention over standard attention in terms of Mean Absolute Error (MAE) for three different time horizons. The relative improvement is obtained by subtracting the final MAE value of MLR attention with the final MAE value of standard attention normalized by the final MAE value of standard attention. We observe that as the sequence length grows, MLR attention outperforms standard attention in oil temperature prediction accuracy.

Finally, we substitute the attention mechanism in the foundational model of Chronos Ansari et al. (2024). Similar to our text experiments, MLR attention reduces the computational cost as seen in Figure 15. For this experiment, we train a Chronos model on the data mixture from Ansari et al. (2024) using a combination of synthetic kernel data and real data. We use an MLR rank split of $32|10|8|8|4|2$.

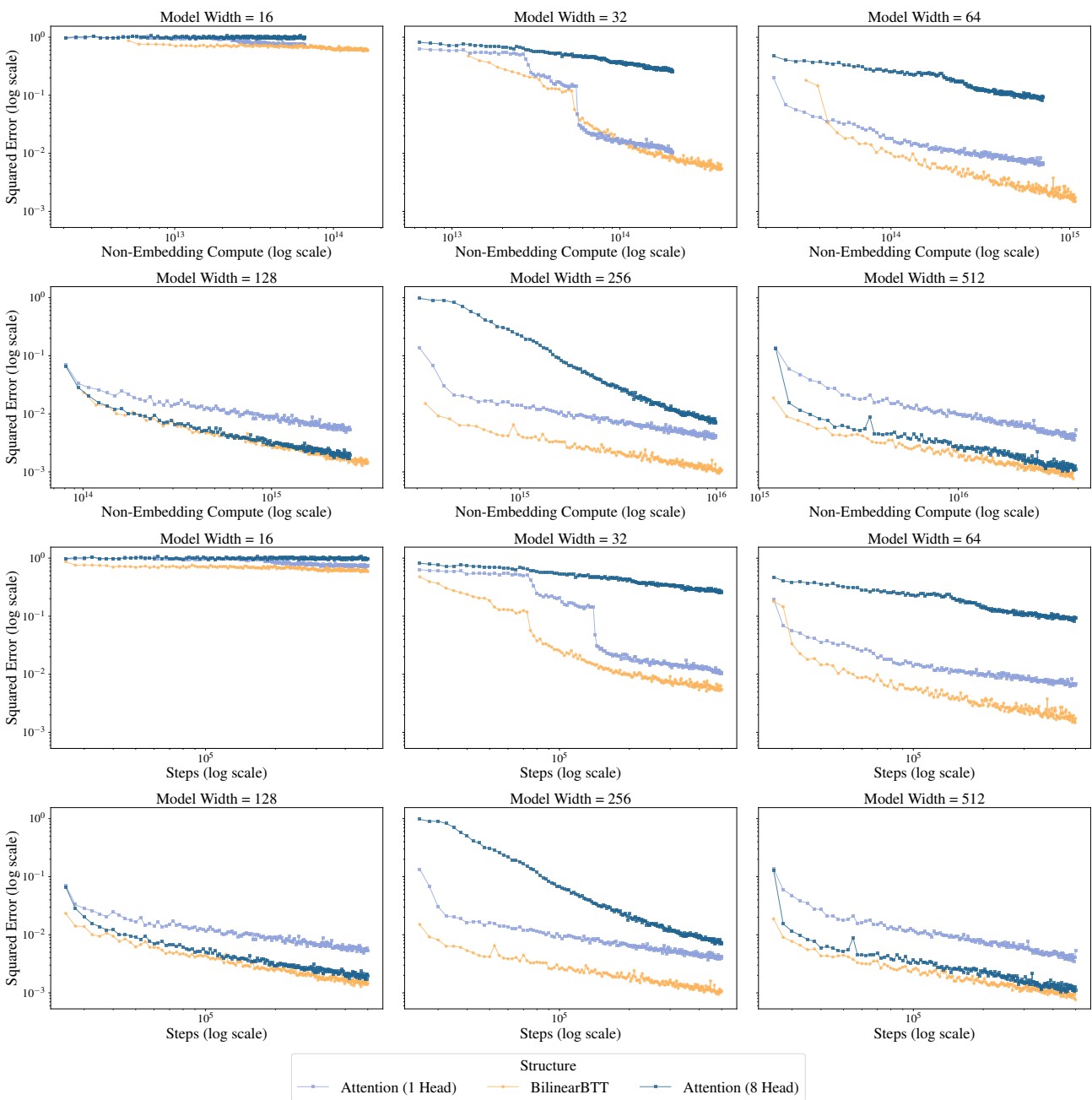

*Figure 7.* **We show in-context regression performances as functions of training steps and training compute (excluding the input and output linear projection layers) across model width** $D \in \{16, 32, 64, 128, 256, 512\}$ **for** $d_{\text{input}} = 16$**.** In general, Bilinear BTT outperforms attention with varying number of heads.

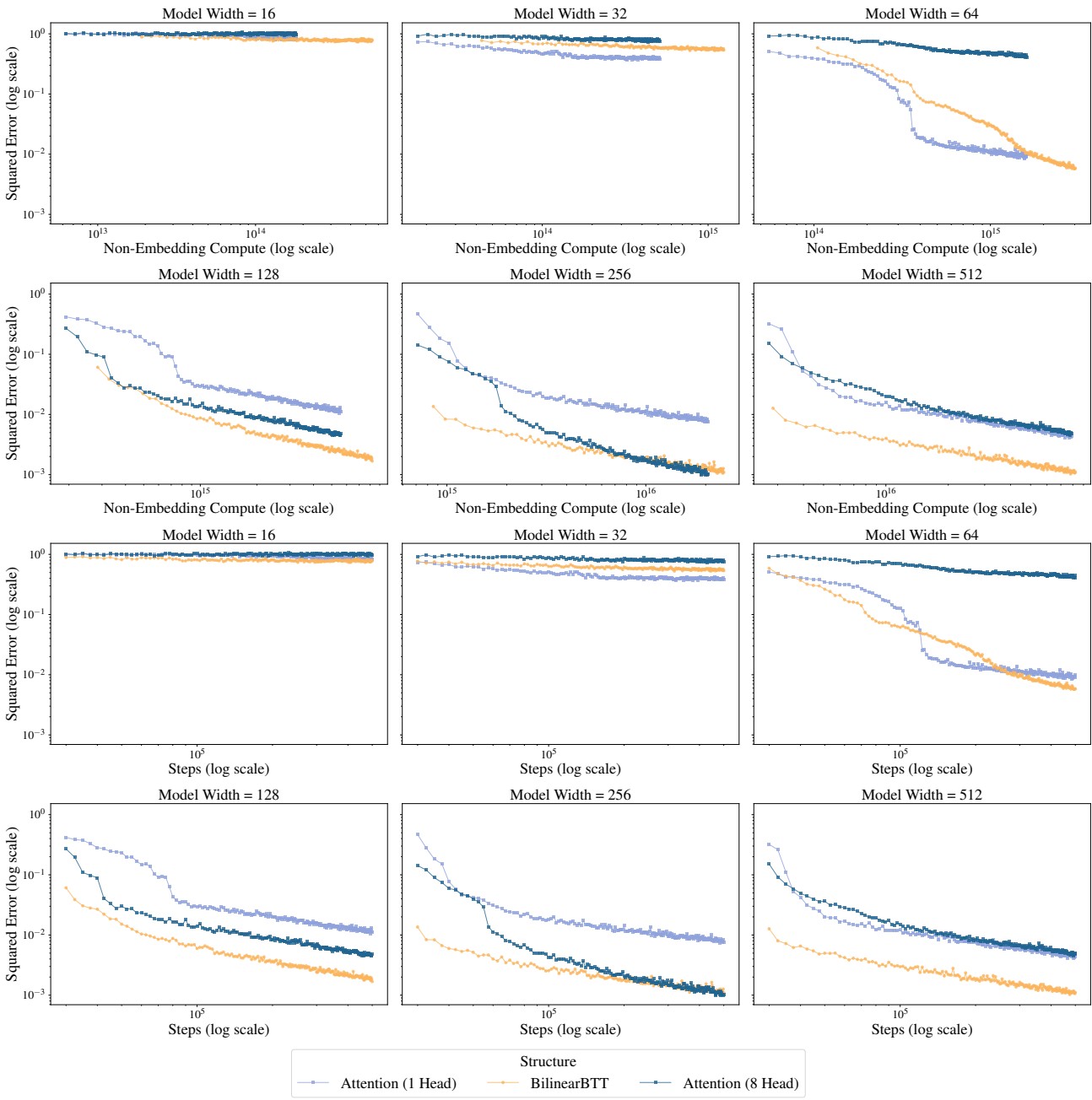

*Figure 8.* **We show in-context regression performances as functions of training steps and training compute (excluding the input and output linear projection layers) across model width** $D \in \{16, 32, 64, 128, 256, 512\}$ **for** $d_{\mathbf{input}} = 32$**.** In general, Bilinear BTT outperforms attention with varying number of heads.

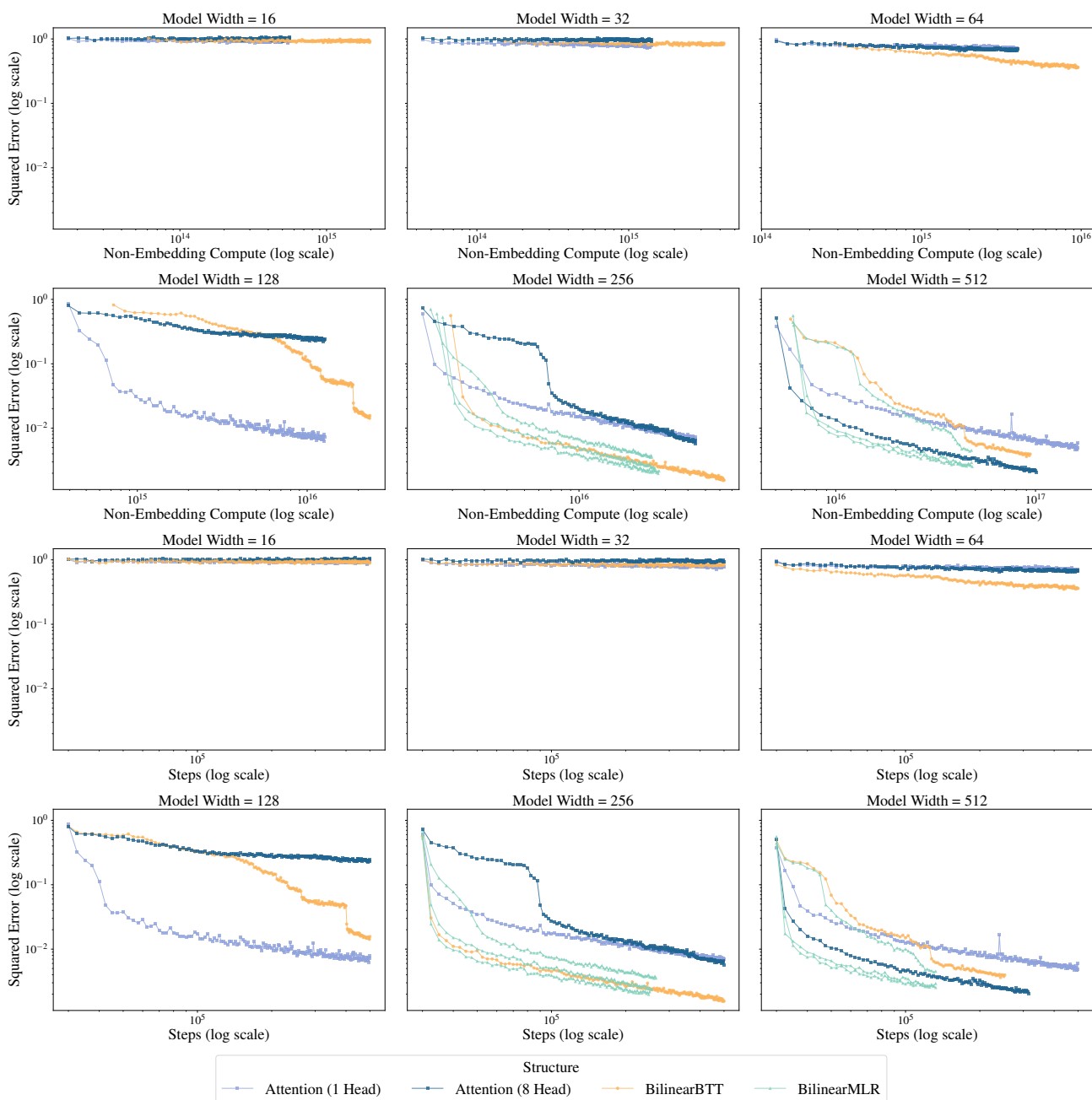

*Figure 9.* **We show in-context regression performances as functions of training steps and training compute (excluding the input and output linear projection layers) across model width** $D \in \{16, 32, 64, 128, 256, 512\}$ **for** $d_{\mathbf{input}} = 64$**.** In general, Bilinear BTT outperforms attention with varying number of heads. Bilinear MLR also outperforms Bilinear BTT when $D = 256$ or $512$. The different Bilinear MLR lines correspond to different 8 levels Bilinear MLR based on $(r_1, \cdots, r_8)$.

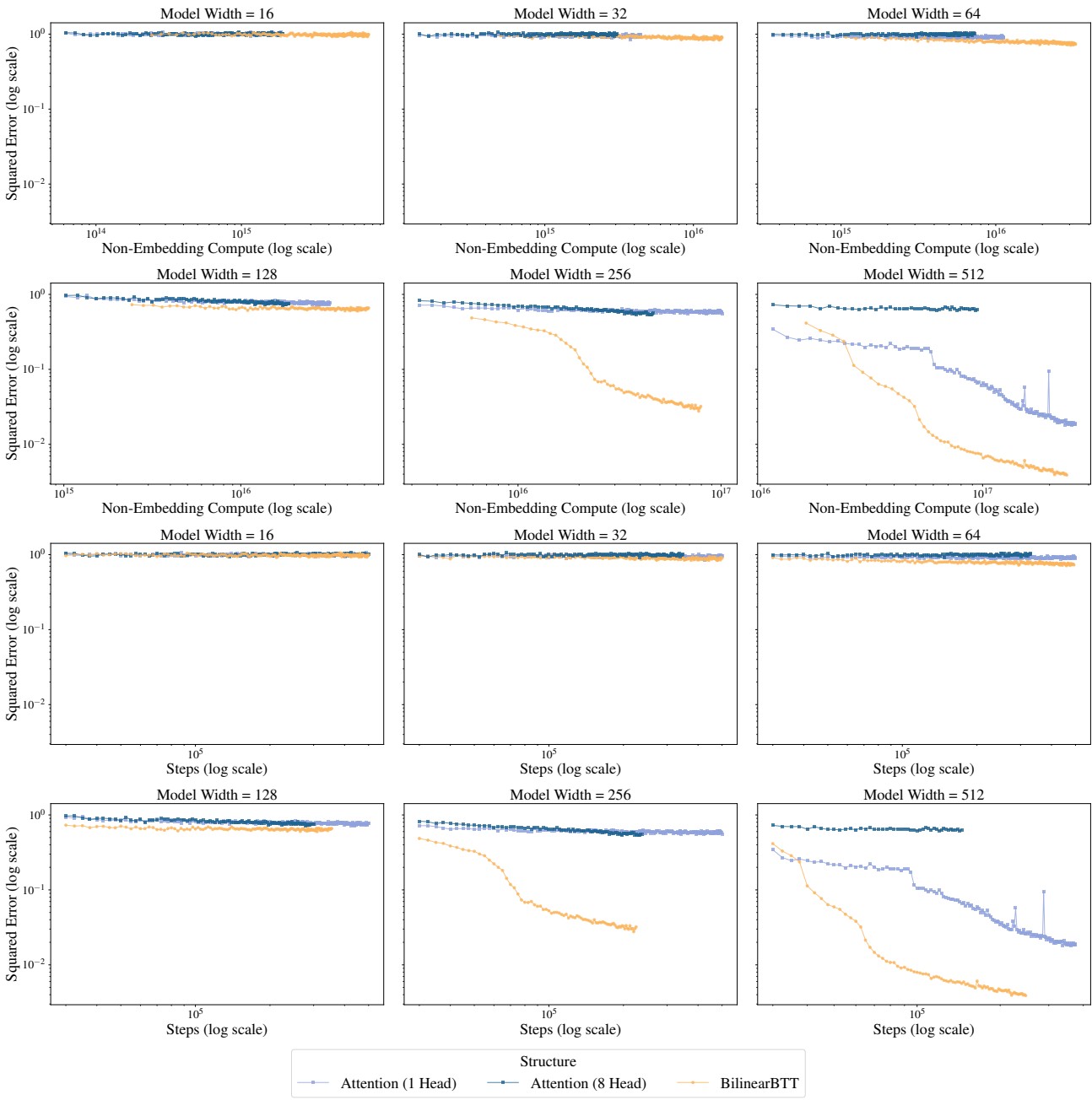

*Figure 10.* **We show in-context regression performances as functions of training steps and training compute (excluding the input and output linear projection layers) across model width** $D \in \{16, 32, 64, 128, 256, 512\}$ **for** $d_{\mathbf{input}} = 128$**.** In general, Bilinear BTT outperforms attention with varying number of heads.

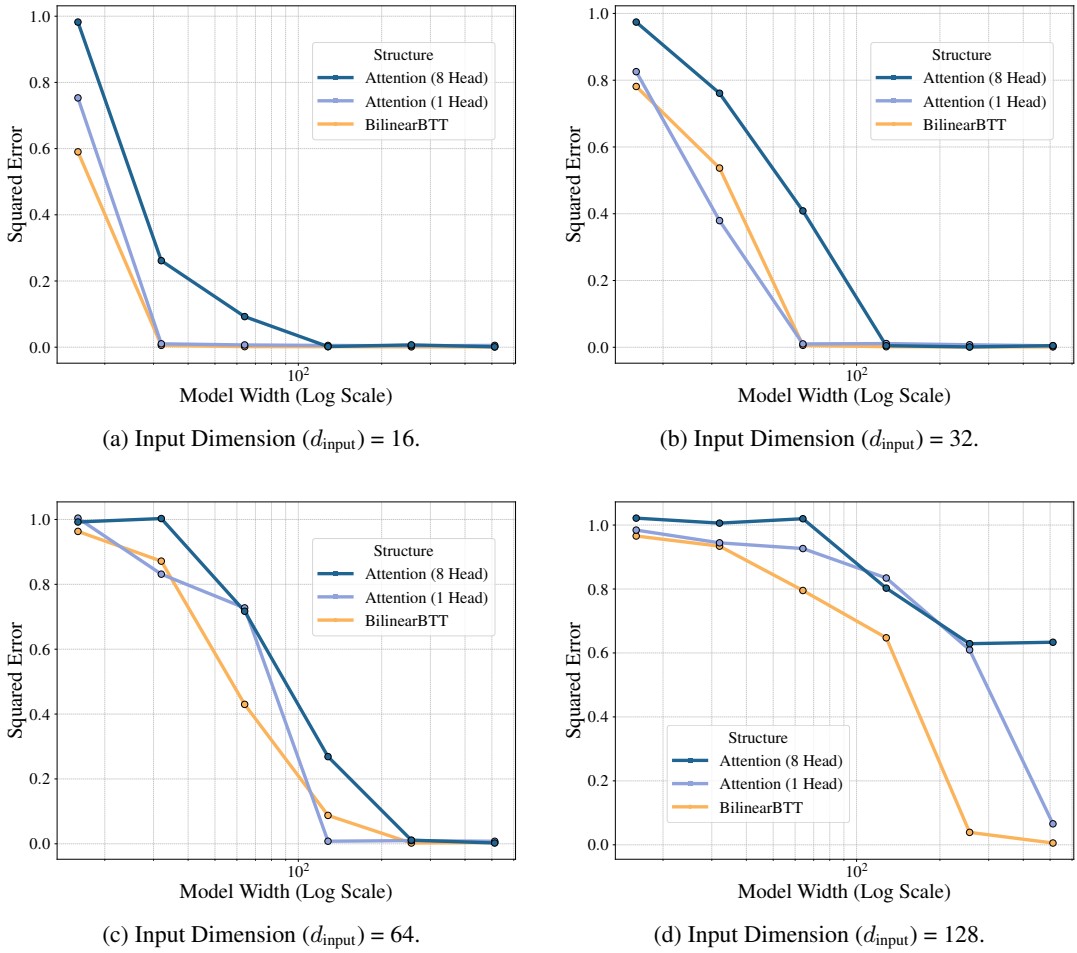

(a) Input Dimension ($d_{\text{input}}$) = 16.

(b) Input Dimension ($d_{\text{input}}$) = 32.

(c) Input Dimension ($d_{\text{input}}$) = 64.

(d) Input Dimension ($d_{\text{input}}$) = 128.

*Figure 11.* **Bilinear BTT and $1$-head attention are full rank structures and thus achieve lower regression errors (y-axis) compared to $8$-head attention at smaller model width (x-axis).** Each one of the four figures correspond to one particular input dimension $d_{\text{input}}$.

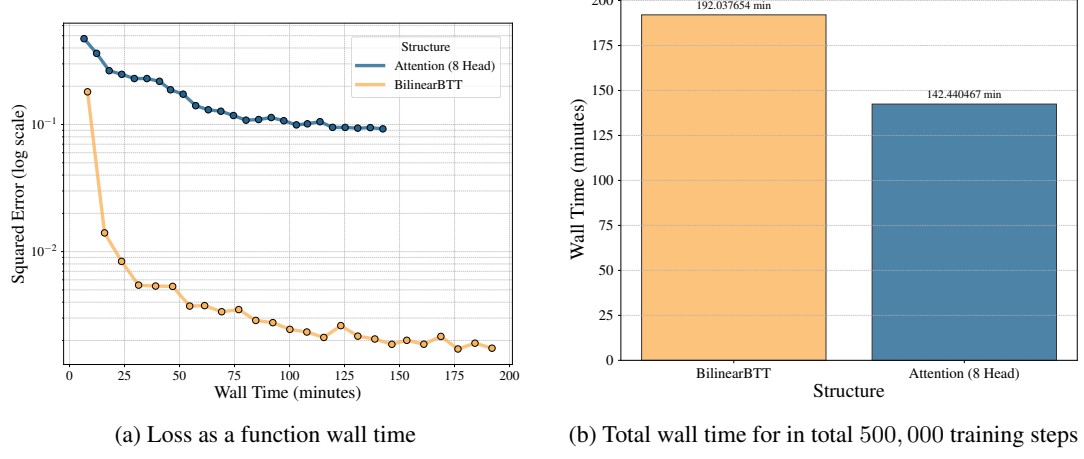

(a) Loss as a function wall time

(b) Total wall time for in total $500{,}000$ training steps

*Figure 12.* **Wall time comparisons between bilinear BTT and standard attention for input dimension at $d_{\text{input}} = 16$ and model width at $D = 64$ and number of attention heads $H = 8$. We omit the $1$ head attention since it has wall time very close to $8$ head attention.** (a) we show the squared regression error in log scale (y-axis) against the wall time between Bilinear BTT and standard attention. We observe that Bilinear BTT still outperforms attention when controlling for wall time. (b) We plot the total wall time for Bilinear BTT and standard attention for $500{,}000$ training steps. Although Bilinear BTT is 1.35x slower, it still has a much better performance compared to standard attention.

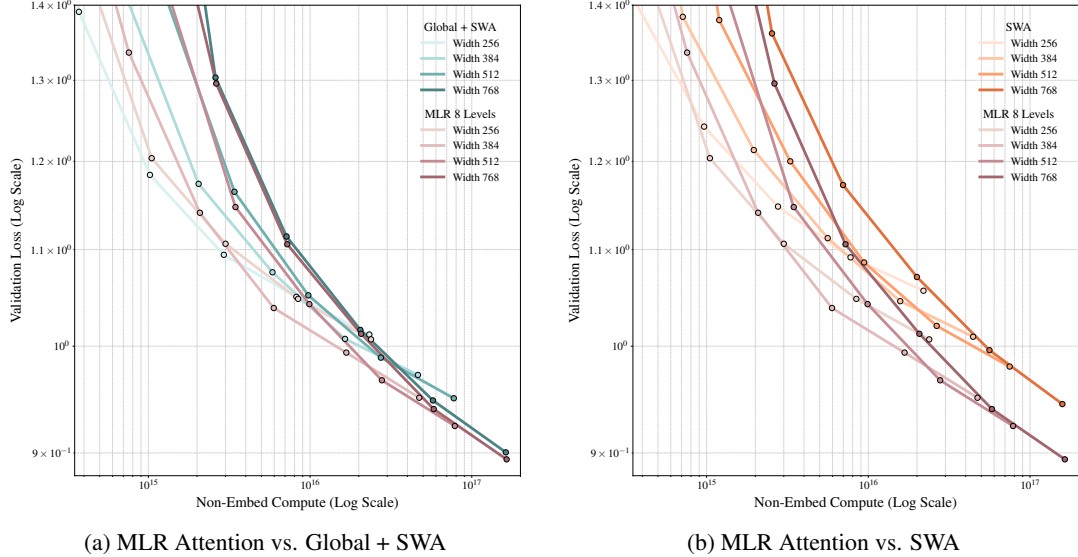

(a) MLR Attention vs. Global + SWA

(b) MLR Attention vs. SWA

*Figure 13.* **MLR attention outperforms both purely sliding window attention (SWA) and combinations of sliding window attention and standard attention with global context across model widths.** SWA refers to all layers being sliding window attention. Global + SWA means sequentially stacking standard attention and SWA.

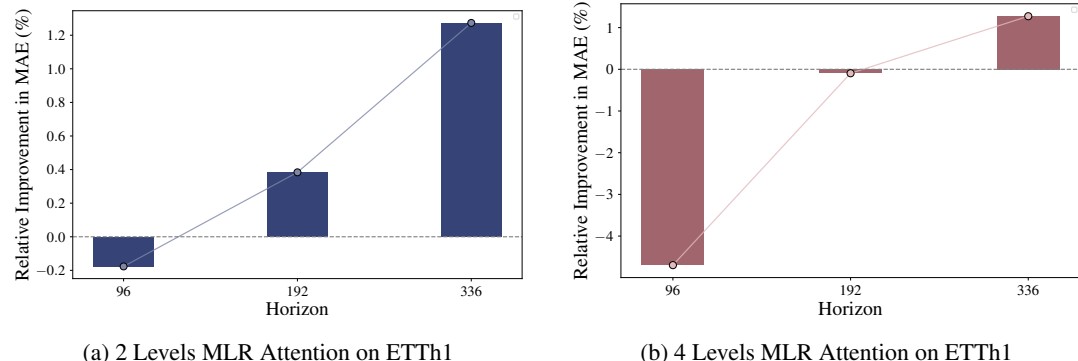

(a) 2 Levels MLR Attention on ETTh1

(b) 4 Levels MLR Attention on ETTh1

*Figure 14.* **As the time horizon (i.e. sequence length) becomes larger, MLR attention outperforms standard attention in oil temperature prediction accuracy by around** $1\%$**.** (a) We show the relative improvement in Mean Absolute Error (MAE) of a 2-levels MLR attention with rank distributions $48|16$. For both horizons 96 and 336, MLR attention achieves better oil temperature forecasting accuracy. (b) We show that a 4-levels MLR attention with rank distributions $40|16|4|4$ outperforms standard attention in relative MAE as the sequence length becomes larger. Across the horizon, we observe a positive correlation between relative MAE and the time horizon of the oil temperature data in the ETTh1 datasets. See Appendix J for additional details.

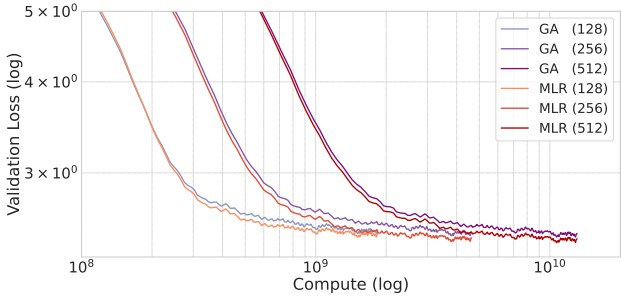

*Figure 15.* **MLR attention reduces computational cost when compared to standard attention.** We plot the validation loss of three distinct model sizes when substituting the T5 attention mechanism in Chronos (Ansari et al., 2024) from standard attention to MLR attention. As shown in the figure, MLR attention achieves the same validation loss compared to standard attention with smaller FLOPs.

