# OpenReview forum: "Customizing the Inductive Biases of Softmax Attention using Structured Matrices"
_ICML.cc/2025/Conference — ICML 2025 poster_

### Official Review · Reviewer_zR4a · 2025-03-04

**Overall Recommendation:** 3

**Summary:**

--- score update to 3 (weak accept) from 2 (weak reject) after the rebuttal ---

The paper consider the Score function in attention computation. This is one core component of modern LLMs. The common setting for multi-head attention is that each head has a "low-rank" bottleneck, as the product of key and query projection matrices ($W_Q W_K^\top$) is a low-rank matrix. Prior work has shown that this precludes learning some functions when doing in-context regression.
Furthermore, regular attention uses the full context which is computationally expensive, relative positions are encoded via positional encodings and are not explicitly treated in the attention itself anymore. An existing proposal to make the attention more efficient is sliding-window attention.

The paper proposes families of structured matrices to parametrize $W_Q W_K^\top$ in ways that are still computationally efficient but lead to full rank matrices.

**Claims And Evidence:**

The paper claims to a/ solve the problem encountered with low-rank approximations for problems that intrinsically require high-rank. b/ improve language modelling tasks by including a locality bias.

The claims are not made very precise throughout the paper (no formal statements) and the empirical evidence seems a bit weak.

Also the claim "attention uses the same scoring function for all input pairs without imposing a locality bias" in the abstract seems misleading to me. Early models used positional encoding with the embedding, whilst Llama models and many others use ROPE in each attention layer. Hence, it is certainly the case that the attention use some relative information about the the tokens.

Overall it is a bit unclear whether they aim to be better (quality) or more efficient. Or pareto-better.

**Essential References Not Discussed:**

see above

**Experimental Designs Or Analyses:**

see above.

**Methods And Evaluation Criteria:**

The benchmark with the in context regression seems a bit limited.

For the language modelling task, I would expect some discussion of the positional encoding.

**Other Comments Or Suggestions:**

Overall, I am somewhat insecure about the impact of the paper. I don't see a large theoretical contribution and the empirical results are a bit limited.

**Other Strengths And Weaknesses:**

--

**Questions For Authors:**

### 1. I am struggling a bit to follow the Locality Bias discussion in section 3.4.

Could you give me a bit more explanation of how Equation 9 get's transformed into sth of the form of equation 4?

In particular (9) requires some explicit handling of the relative indices, whereas 4 does not.

Also in (9) my understanding is that there is a maximum "reach" defined by the maximum level. This would mean that it cannot consume the full context and is actually more like a sliding window.

---
### Figure 3:
Am I understanding correctly that Attention with 1Head uses the full rank $D \times D$ matrix? And yours is always using a single head as well, right?
From an expressivity perspective, 1 Head Attention should be strictly better that BTT. Are you hence making a claim about training dynamics?? Also in Appendix Figure 9, the ordering seems not consistent. How did you choose to report D=128 in the main document?

**Relation To Broader Scientific Literature:**

As mentioned above I am missing a discussion of positional encodings. Furthermore, modern LLMs often use GroupedQueryAttention. I think for larger impact those should be discussed.

**Theoretical Claims:**

The only theoretical claim is that MLBTC (Definition 1) subsumes MLR and BTT matrices. This seems correct.

---

> ### Author Rebuttal · Authors · 2025-04-01
>
> Thank you for your constructive feedback. We respond to your questions below.
>
> **Relationship with Positional Encoding and RoPE.**
>
> It is certainly true that many modern transformers use positional encoding schemes that capture relative position. For instance, RoPE encodes positional information by rotating query and key vectors by an angle proportional to the token’s index in the sequence. This can help the scoring function treat neighboring tokens alike. So in a way, RoPE does have a “locality bias”, or at least a "locality awareness".
>
> But the goal of our MLR attention is different. We are not trying to simply encode relative positional information or boost the attention score between neighboring tokens. Instead, we change the computational cost of the scoring function based on the tokens’ positions. Put differently, standard attention (with or without RoPE) uses the same query/key dimension for all pairs of tokens. We effectively use a smaller query/key dimension for tokens that are far apart. This saves FLOPs compared to standard attention, while still allowing high quality attention scores for neighboring tokens. Thus our MLR attention is compatible and complementary with existing positional embedding schemes, including RoPE.
>
> We acknowledge that our choice of the term “locality bias” in the context of MLR attention may have obscured our meaning. In our revised manuscript, we instead call it "distance-dependent compute allocation".
>
> **Do we "aim to be better (quality) or more efficient"?**
> 1) Bilinear BTT improves expressive power by increasing the effective query/key dimension of each head (“the rank”). However, it costs more parameters and compute than standard attention (see Table 1). That is, it increases “quality” at the possible expense of efficiency. Figure 3b shows that this trade is worth it.
> 2) MLR attention improves efficiency by decreasing the effective query/key dimension for pairs of tokens that are far apart in the sequence. That is, it saves FLOPs at the expense of expressiveness. Because real-world sequence data has local structure, this reduction in expressive power probably does not matter.
>
> **Grouped Query Attention (GQA).**
> In GQA, the same KV transformation is shared across several heads. Our methods can be thought of in terms of query and key transformations too; see Appendix B.2 and the discussion under Eq 9. Thus, our approach is fully compatible with GQA. We include a detailed discussion of GQA in our revision.
>
> **Explanation of how Equation 9 gets transformed into S of the form of equation 4.**
> Claim: The $j, j’$ entry of S equals the right hand side of Eq 9.
> Proof. For simplicity, assume there are just two levels, so Eq 9 reduces to the formula preceding it on line 246.
> Divide $X = \begin{bmatrix}X_1 \\\\ X_2\end{bmatrix}$ into blocks. $X_1$ is the first half of the sequence and $X_2$ corresponds to the second half. Divide $W_Q = \begin{bmatrix}L_1 & L_2\end{bmatrix}$ (line 270). Thus $$Q = X W_Q = \begin{bmatrix}X_1 L_1 & X_1 L_2 \\\\ X_2 L_1 & X_2 L_2\end{bmatrix}$$
> Now divide $Q$ into 3 named blocks according to Figure 1b:
> - $Q_{11} = XL_1$
> - $Q_{21} = X_1 L_2$
> - $Q_{22} = X_2 L_2$
>
> Analogously
> - $K_{11} = XR_1$
> - $K_{21} = X_1 R_2$
> - $K_{22} = X_2 R_2$
>
> Finally, plug the above into the definition of S on line 268:
> $$S = Q_{11}K_{11}^\top + \begin{bmatrix}Q_{21}K_{21}^\top & 0 \\\\ 0 & Q_{22} K_{22}^\top\end{bmatrix} = XL_1R_1^\top X^\top + \begin{bmatrix}X_1 L_2 R_2 ^\top X_1^\top & 0 \\\\ 0 &X_2 L_2 R_2^\top X_2^\top\end{bmatrix}$$
>
>
> Consider the $j, j'$ entry of $S$. If they’re in different blocks, then it’s $x_j^\top L_1 R_1^\top x_j$. If they're in the same block, it's $x_j^\top L_1 R_1^\top x_j + x_j^\top L_2 R_2^\top x_j$.
>
> We added this derivation to the appendix.
>
> **Is there a maximum "reach" defined by the maximum level?**
>
> The first level is global, so all tokens interact. In the notation of Eq 9, this is because *all* pairs of tokens $j, j’$ have $d(j, j’) \geq 1$.
>
> **Does Attention with 1 Head use the full rank D×D matrix?**
>
> Yes.
>
> **Does our attention always use a single head as well?**
>
> We use multiple heads. E.g., in Fig 3, our bilinear models use 8 heads.
>
> **How did we choose to report D=128?**
>
> In all four subplots of Figure 9, standard 1-head attention and BilinearBTT (both full rank) converge well before standard H=8 attention (which is low rank). This is the only claim we make with this figure or with Fig 3a. We picked d=128 because it was the largest and most realistic. In Fig 3b, we further show that Bilinear BTT is better than 1-head attention because it trains faster in terms of FLOPs.
>
> Thank you again for your detailed feedback and your questions. We made a significant effort to address your questions, and we would appreciate it if you would consider raising your score in light of our response. Do you have any additional questions we can address?

---

### Official Review · Reviewer_z3b6 · 2025-03-10

**Overall Recommendation:** 4

**Summary:**

--- increased score from 3 to 4 after comment from authors ---

This work proposes a new way to parameterize the query-key operation in attention matrices. When the key and query matrices are low-rank, as it is the case in the vanilla transformer architecture, high-dimensional input data might suffer from a low-rank bottleneck leading to low performance. While introducing full-rank matrices would solve this problem, this increases the number of flops needed to run the model. Therefore the authors propose a new version of parameterizing this operation via highly structured high-rank matrices. This not only increases efficiency compared to the dense matrices and allows representing high-rank operations, but also allows taking in distance between tokens as a parameter to model locality.

**Claims And Evidence:**

The claims on the parameterizations themselves are valid, i.e. how they relate to one another and their parameter counts.

The evidence that the new parameterizations are more efficient is shown in experiments on language, in context learning and regression. However, it is unclear from the manuscript how the training time, hyperparameter search and final performance, as well as the number of heads influence one another. For practicioners, it would be useful to obtain a deeper understanding of there links, to be able to decide in which cases to use this architectural component specifically.

**Essential References Not Discussed:**

Two factors that might be discussed more in depth are the following:

(Q1) Since the authors mainly advocate that their approach is more efficient, it would be useful to know how exactly the flops and "non-embedding compute" are calculated. Also it would be useful to know how this approach impacts parallelization. Similarly, when the authors mention on page 5, "DxD matrix is prohibitively expansive", I would like to understand better why. After all the rule of thumb is that the hidden layers have size 4D - so the computation there is even more expensive. What is the difference between the two settings that motivates your statement?

(Q2) The statements about locality never consider the fact that most input data is decorated with positional information via positional embedddings. It seems that this already adds a sort of positional bias to the matrix when the positional embeddings are relational and their outer product represents a structured matrix. How does this relate to your approach?

(Q3) You mention on page 4 that the BTT matrix has been used as a replacement for linear layers in neural networks. This is elaborated on in section 6, first paragraph. Can you make the special properties of the query-key-matrix more explict, that warrant a special examination, rather than simply treating it as a linear layer and replacing it with those previous methods?

(Q4) Is there work that investigates the trade-off between the number of heads and the attention rank?This would seem related to the efficiency discussion, as well as your experiments where you compare 1- and 8-head models. Some known mechanisms, such as induction heads [1] expllicitly use several heads to execute different functions, it is unclear how your proposal would be able to implement them.

[1]Olsson, Catherine, et al. "In-context learning and induction heads." arXiv preprint arXiv:2209.11895 (2022).

**Experimental Designs Or Analyses:**

It would be useful to include error bars in all experimental plots over several runs to estimate the variance and significance of the differences.

**Methods And Evaluation Criteria:**

see above.

**Other Comments Or Suggestions:**

The figure lables are quite small compared to the text.

Even though at this point in time some elements of the introduction seem like universal truths, it would be correct to cite the appropriate related work when mentioned, i.e.
- L.43: the attention mechanism/transformer
- L.47: transformers are being used as general purpose tools
- L.48: Transformers have specific inductive biases
- L.14: a large ongoing research effort ... for long and big models
- L.34: lacks a bias for prioritising local interactions - how is this related to positional encodings? - why is the locality bias from them not enough?

**Other Strengths And Weaknesses:**

.

**Questions For Authors:**

See all questions (QX) before in the relevant sections.

(Q5) On page 5, left first paragraph you choose $s=1$ or $s=2$. I might have misunderstood the previous section, but does this imply that the largest matrix you can fully represent for $s=1$ is a 1x1 matrix? Could you elaborate on this together with the practical values you choose in your experiments? Or is there a typo in the $a=b=c=d=s=\sqrt{d}$ constraint?

(Q6) Why do the 8- and 1-head attention structures in Fig.3b) have the same points on the x-axis? Should the 8-head one not be more compute intense? Maybe it woud be helpful to clarify which settings you are using here in the caption.

(Q7) In Figure 3a), what do you mean by "controlling for the number of training steps"?

(Q8) This might be me, but can I use several heads of MLR and BTT, or combine them? Would it make sense to compare the 8-head attention to the 8-head setting of the structured matrices?

The responses to the questions here would improve the presentation and understanding of the paper, specifically to make the efficiency improvements more clear empirically - this would strengthen the main claims of the paper.

**Relation To Broader Scientific Literature:**

The work examines attention mechanism closely, which is itself being studied widely due to the high frcation time and compute constraints it ensues. Next to the other models of decreasing computational efficiency this is another valid and interesting approach.

**Theoretical Claims:**

There is no strong theoretical claims except the parameter efficency that are to check, the inclusion of the related matrix families seem ok.

---

> ### Author Rebuttal · Authors · 2025-04-01
>
> Thank you for your thoughtful questions and supportive feedback!
>
> **Hyperparameter and Error Bars.**
>
> In all experiments, for both our methods and baselines, we sweep over the learning rate while keeping other hyperparameters fixed. In all figures, including Fig 3 and Fig 4, we plot only the best learning rate for visual clarity. Since our code derives from the NanoGPT codebase, the other hyperparameters (weight decay, Adam betas, etc.) are well-tuned for standard attention already. We did not have the compute budget to sweep multiple random seeds for our larger experiments, though we can add error bars for the smaller-scale experiments to our revision.
>
> The choice of learning rate is further discussed in Appendix E.1. Our implementation adopts muP, a method for stable hyperparameter transfer across model width. Please refer to the “Larger Models and Datasets” section in our response to Reviewer 1ojm for a figure that plots loss against various learning rates. We hope that this helps address your concern that we indeed pick the best learning rate for both the baseline and our method.
>
> **Response to Review’s Questions.**
>
> ### Q1
> You can refer to the `compute_model_FLOPs` function in `src/utils.py` provided in our codebase. We use FlopCountAnalysis from the fvcore library. This function traces the forward pass of the model and records the total number of FLOPs for all the operations. To compute the non-embedding FLOPs, we subtract the FLOP count of the token embedding and language modeling head (the first and last linear layer)
>
> The MLP layers have weight matrices of size 4D x D, but the attention layers have many heads. It would be too expensive for *each* head to have a D x D weight matrix. For example, Llama 3, 70B has H = 64 heads, D = 8192, and 80 layers. If each head used an unstructured DxD weight matrix instead of the D x (D/H) matrices $W_Q$ and $W_K$, it would have 330 billion more parameters.
>
> ### Q2:
> We kindly refer you to our response to Reviewer zR4a.
>
> ### Q3:
> Previous methods were for linear layers of the form $X \mapsto AX$. We use it for $x \mapsto X W_Q W_K^\top X^\top$. That is, previous work on replacing linear layers with structured BTT could not account for the fact that a structured matrix is *already* being used in standard attention, since $W_Q$ and $W_K$ function not as two separate linear transformations, but a single, low-rank (bi-)linear transformation.
>
> ### Q4:
> - Like standard attention, our proposed methods use multiple attention heads in each attention layer. Our modifications to standard attention are applied separately to each head. In Fig 3, we compare our method to standard attention with 1 head as a baseline since 1-head attention is full rank like our version, but that isn’t our proposed method. We don’t explicitly test for induction heads, but we think our methods can implement them as well as standard attention, especially since they succeed at language modeling and in-context linear regression.
> - There is some work on the tradeoff between number of heads and rank. Theoretical: Amsel (https://iclr.cc/virtual/2025/poster/27747) and Sanford, “Representational Strengths and Limitations of Transformers” (https://openreview.net/pdf?id=36DxONZ9bA). Empirical: Bhojanapalli (https://dl.acm.org/doi/10.5555/3524938.3525019) and Appendices D.4 and E.2 of the muP paper https://arxiv.org/pdf/2203.03466.
>
> ### Q5
> Let us fix $a=b=c=d=\sqrt{D}$, as in our experiments. We are now free to set s to anything between 1 and $\sqrt{D}$ to trade off efficiency for expressivity. (Because of the cited result, setting $s > \sqrt{D}$ would be pointless.) We chose to set s = 1 or 2 to maximize speed, and we find that the model is still expressive enough to perform well (even better than low rank attention). In fact, when s=1, BTT matrix is exactly a Monarch matrix (https://arxiv.org/abs/2204.00595), which is an expressive matrix class already.
>
> ### Q6
> The 8-head and 1-head models we test have exactly the same compute intensity. This is because the 1-head model has an 8x larger head dimension, due to the Hr=d rule.
>
> ### Q7
> We mean to say that each point on this graph corresponds to a model that was trained for the same number of steps. We have now clarified this point in the caption.
>
> ### Q8
> We are currently using multiple heads of BTT or MLR attention. We generally compare our attention to standard attention with the same number of heads. In Fig 3, we additionally compare to standard attention with one head. We explain this more clearly in the revision.
>
> **Writing and Style**
>
> Thank you for your suggestions about enlarging figure labels and including citing some standard claims. We have incorporated them in our revised manuscript.
>
> We made a significant effort to address your questions, including paper edits which we feel improve the clarity of our paper. We would appreciate it if you would consider raising your score in light of our response. Do you have any additional questions we can address?

---

> > ### Comment · Reviewer_z3b6 · 2025-04-02
> >
> > Dear authors,
> > Thank you for the explanations, that improved my understanding of the paper, and hopefully will help future readers as well in the future after you edited them in. In agreement with reviewer zR4a I think the choice of "distance-dependent compute" is very helpful to the intuitive understanding. I also apprechiate that you will add the error bars for smaller scale experiments, but have complete understanding that with limited compute budget it is not possible to do it for the larger scale experiments.
> > Best!

---

> > > ### Author Response · Authors · 2025-04-08
> > >
> > > We are grateful to the reviewer for the constructive feedback! We will add the explanations and error bars for smaller scale experiments in our revision. We will also change the term locality bias to distance-dependent compute. Thanks to the Reviewers' comments, these will undoubtedly improve the manuscript.

---

### Official Review · Reviewer_1ojm · 2025-03-10

**Overall Recommendation:** 4

**Summary:**

The authors propose to address a common limitation of existing softmax attention layers: the information bottleneck when using small head dimension. To do this, they propose to bake in a locality bias into the structured parameterisation of the attention weights.
The attention mechanism is introduced cleanly as a billinear form, from which they can introduce structured and parameter efficient bilinear forms. This work appears to build upon the work of Parshakova et al but in the context of designing an efficient attention layer.

**Claims And Evidence:**

The main claim is that current attention layers have an information bottleneck, which is more significant when the head dimension is reduced. Naturally, this depends on the data, but the intuition is valid, the results confirm this, and the authors provide significant references to existing works that highlight both the theoretical and practical cases where this can be a problem.

Secondly, they propose that a locality bias is a good way to introduce more parameter efficiency, without degrading performance. The authors confirm this with emperical results.

**Essential References Not Discussed:**

Other than [1], the discussions are relatively complete.
[1] Rethinking Attention with Performers. ICLR 2021

**Ethical Review Flag:**

Flag this paper for an ethics review.

**Experimental Designs Or Analyses:**

The wall clock time is great to see and strengthens the practical impact of this paper. The implementation using batch matrix multiplication is simple and effective for these structured block matrices.

**Methods And Evaluation Criteria:**

The authors evaluate on the OpenWebText dataset and the ETT time series forecasting dataset, where they observe better performance as the sequence length grows. Although the first dataset is moderate size, a large scale dataset and larger models would give this paper a much bigger impact. Although understandably, there are new difficulties when scaling up these methods to ensure theoretical flops does translate to real wall clock performance reductions.

**Other Comments Or Suggestions:**

None

**Other Strengths And Weaknesses:**

One of the main limitations is not seeing more quantitative results, larger models etc.

**Questions For Authors:**

None

**Relation To Broader Scientific Literature:**

This work naturally builds upon existing and recent development of multi-level low-rank matrices. These fit very nicely into the attention matrix formulation and are well motivated. This work has a very broad impact, and further practical developments on efficient implementations could enable its widespread adoption among practitioners.

**Theoretical Claims:**

I looked through the derivations and they appear to be correct.

---

> ### Author Rebuttal · Authors · 2025-04-01
>
> We thank you for your supportive feedback. We address your comments below.
>
> **Larger Models and Datasets:**
>
> We are hopeful about the prospect of scaling up these experiments to even larger models and datasets. As a first step in that direction, we now present a new experiment on hyperparameter transfer for our architectures. The maximum update parameterization (muP) has become a crucial tool for scaling models up because it provides a way to transfer the results of hyperparameter tuning from small models to large ones. This is far less expensive than tuning hyperparameters for large models directly. For architectures that use structured matrices, muP does not work out of the box, but in Appendix E of our paper, we provide a recipe for adapting muP to our architectures. Our latest experiment validates this recipe.
>
> In the figure provided in the link https://drive.google.com/file/d/18pI--DmWWqB3PqFd-dEaziOypc5rRIP8/view?usp=sharing, we show the validation loss of an 8 Level MLR attention and standard attention on OpenWebText across a variety of learning rates. In our paper, the maximum model width we used is 768 with a context length of 1024. Here we sweep over model widths  512, 768, and 1024 with a reduced context length of 256 due to compute constraints. As the figure shows, our MLR attention shares the same optimal learning rate across model width, and it’s also consistently better than standard attention when both are properly tuned. This result suggests MLR will continue to perform well on larger models trained with more data.
>
> **Efficient Implementation.**
>
> As the review notes, a thoughtful IO-aware implementation is needed to ensure that savings in FLOPs translate to savings in wall-clock time on a GPU. Like standard attention, our proposed methods rely on a few batch matrix multiplications followed by softmax. The block diagonal structure of MLR matrices is easy to parallelize, and we think the highly structured permutation matrix in BTT can be fused with the surrounding multiplications. Thus, we are optimistic that techniques similar to Flash Attention can be applied to our attention variants (and indeed, to general MLBTC matrices).
>
> **Rethinking Attention with Performers. ICLR 2021**
>
> Thank you for pointing out this paper! We have incorporated a discussion of it in the revised manuscript.
>
> We would like to note that while the Performers paper proposes an algorithm for computing attention more efficiently achieving a linear instead of quadratic dependence on the sequence length, the function they are (approximately) computing is the standard attention function. As a result, they still have a low rank bottleneck and they lack a locality bias in the allocation of compute. In contrast, our approach to improving transformer models is to replace the standard attention function with something else. Our proposed methods have different inductive biases from standard attention, which we show leads to better performance in several cases. However, we retain the quadratic dependence on the sequence length. We leave it to future work to find efficient algorithms for (approximately) implementing our proposed architectures.
>
> Thank you again for your detailed and supportive review. We hope we addressed your questions. Please let us know if you have any additional questions or comments that we can discuss.

---

> > ### Comment · Reviewer_1ojm · 2025-04-04
> >
> > I appreciate the authors comments in addressing the discussion with Performers. The comparison is good to see and should be included in the main manuscript. Furthermore, the additional results scaling to a larger model is great to see! I maintain my original score.

---

> > > ### Author Response · Authors · 2025-04-08
> > >
> > > We are grateful to the reviewer for the constructive feedback! We will add the new figures, discussions, and references. Thanks to the Reviewers' comments, these will undoubtedly improve the manuscript.

---

### Official Review · Reviewer_g9Ct · 2025-03-14

**Overall Recommendation:** 3

**Summary:**

The paper address two issues of the attention computation.

The first is the bottleneck caused by the low rank computation of the Key and Query matrices in the attention computation. Instead of doing the standard low rank decomposition, they proposed to use structure matrices to represent the attention score, which is full rank but still more efficient in computation that the full DxD matrix (O(D^(2/3)) v.s. O(D^2)).

The second is not paying more attention to locality. Most problems have locality properties where the closer tokens are often more relevant than the further tokens. By introducing structure matrices, the proposed method introduce locality into attention computation.

**Claims And Evidence:**

* Better at the efficiency and performance trade-off.
  - Figure 10 in the Appendix shows the loss as a function of wall time.

* Locality.
  - The paper includes experiment on regression problems where the standard attention performs poorly and show that the proposed method can improve the performance (Figure 3).

**Essential References Not Discussed:**

N/A.

**Experimental Designs Or Analyses:**

The paper uses experiments measuring squared error on in-context regression tasks and shows that the proposed method outperforms the standard method.

**Methods And Evaluation Criteria:**

Yes. The proposed methods introduce structure into the attention computation that was originally represented as low-rank matrix multiplication. The method is evaluated on in-context regression, which highlight the low-rank bottleneck issue.

**Other Comments Or Suggestions:**

Typos:
  - line 033. Section 1, paragraph 2, third last line. "express with attention".
  - line 304. Section 4, paragraph 3, last line. The equation is missing a "(".

**Other Strengths And Weaknesses:**

The paper is clear in the target problem and proposed methods to solve the problem.

The writing of the paper can be improved.

**Questions For Authors:**

I'm wondering if there are textual problems (non-regression problems) where the locality can have an impact.

**Relation To Broader Scientific Literature:**

The paper is proposing structured matrices in attention computation. There were other previous works using different design of structured matrices.

**Theoretical Claims:**

N/A.

---

> ### Author Rebuttal · Authors · 2025-04-01
>
> Thank you for your thoughtful and positive feedback. We address your questions below.
>
> **Contributions in Comparison to Prior Works.**
>
> Thank you for pointing out the connection of our work to the broader literature of structured matrices. We would like to highlight further connections that we did not mention explicitly in the paper:
> - Several prior works [2, 3, 5] replace linear layers in the Transformer model for improved efficiency and a better scaling law. Our work goes beyond them by 1) considering structured matrices in bilinear transformations, like the one that defines the attention score, and 2) using structured matrices to replace data-dependent matrices like QK^T, where the query and key matrices are themselves the outputs of a linear layer.
> - The Multi-Level Low Rank (MLR) matrix [1] was originally proposed in the applied math literature as an extension of low rank matrices to fit multilevel factor models. To our best knowledge, our work is the first to apply MLR matrices to deep learning models.
> - We propose Multi-Level Block Tensor Contraction (MLBTC), a family of structured matrices that generalizes many common structured matrices like Low Rank, Kronecker, Monarch [2], Block Tensor Train [3], Multi-Level Low Rank [1]. We believe this is a novel perspective. We conduct experiments to explore the various inductive biases that these structured matrices encode when applied to a Transformer architecture.
> - The Hydra paper proposed the matrix mixer framework that unifies common sequence mixer modules (e.g. CNN <-> Toeplitz, Linear Attention <-> Low Rank, Mamba <-> Semi-Separable, etc.) with their underlying structured matrices. We believe our efforts add to this line of work of exploring novel structured matrices for sequence mixing.
>
> **Writing.**
>
> Thank you for your helpful feedback regarding the writing and typos. We have updated our manuscript accordingly, alongside additional clarifications inspired by your questions.
>
>
> **Locality.**
>
>  We believe that problems with long context lengths and a hierarchical structure are most amenable to our approach. The computational savings of MLR attention compared to standard attention increases with the sequence length. And the multi-level nature of MLR matrices make them well-suited to hierarchically structured data, where tokens become gradually more and more related as their distance decreases. One application we are excited about is code models. Code repositories tend to be large, pushing the limits of standard transformers' context windows. They are organized hierarchically into folders and files, and each code file is further organized by an Abstract Syntax Tree, with hierarchically nested classes, methods, control structures (loops), lines, and expressions. Our method may help transformers read repositories more efficiently, like humans do, focusing most of their effort on the immediate context, but keeping the global structure in mind too.
>
> We are also intrigued by potential applications to non-text data, such as DNA sequences, phylogenetic data, time series data, and graph data. We plan to apply our method to some of these settings in future work.
>
>
> Thank you again for your detailed review. We hope we were able to address all of your questions and that you would consider raising your score. Please let us know if you have any additional questions or comments that we can address or discuss.
>
> ________
> **Reference:**
>
> [1] Factor Fitting, Rank Allocation, and Partitioning in Multilevel Low Rank Matrices. https://arxiv.org/abs/2310.19214
>
> [2] Monarch: Expressive Structured Matrices for Efficient and Accurate Training. https://arxiv.org/abs/2204.00595
>
> [3] Compute Better Spent: Replacing Dense Layers with Structured Matrices. https://arxiv.org/abs/2406.06248
>
> [4] Hydra: Bidirectional State Space Models Through Generalized Matrix Mixers. https://arxiv.org/abs/2407.09941
>
> [5] Searching for Efficient Linear Layers over a Continuous Space of Structured Matrices. https://arxiv.org/abs/2410.02117

---

### Decision · Program_Chairs · 2025-05-01

**Decision:**

Accept (poster)

**Comment:**

The paper proposes a method for inducing different inductive biases in attention-based models by designing attention alternatives with structured families matrices. It targets two perceived weaknesses of standard attention - the "low rank bottleneck" caused by low head dimension used as inner dimension, and the uniform allocation of computational resources to pairs of tokens irrespective of their distance in the sequences. It addresses them with two families of structured matrices, block tensor train (BTT) and multi-level low rank (MLR).

Reviewers found the problem and approach interesting and well-motivated. The empirical results were a focus of some criticism, as the experiments were deemed only moderate in scale, and the reported performance measures were only combinatorial (FLOP counts) rather than wall-clock times. This leaves some lingering doubts about the viability of the method in practice and its potential impact. However, the innovation of the paper and the preliminary empirical results were deemed sufficiently strong by the reviewers to accept the paper.

Several points were clarified and expanded in the rebuttals, which the authors are encouraged to include in the final version.